# Compositional–ARC: Assessing Systematic Generalization in Abstract Spatial Reasoning

**Philipp Mondorf**[1,2]    **Shijia Zhou**[1,2]    **Monica Riedler**[1,2]    **Barbara Plank**[1,2]

[1]MaiNLP, Center for Information and Language Processing, LMU Munich, Germany
[2]Munich Center for Machine Learning (MCML), Munich, Germany

`{p.mondorf, zhou.shijia, monica.riedler, b.plank}@lmu.de`

## Abstract

Systematic generalization refers to the capacity to understand and generate novel combinations from known components. Despite recent progress by large language models (LLMs) across various domains, these models often fail to extend their knowledge to novel compositional scenarios, revealing notable limitations in systematic generalization. There has been an ongoing debate about whether neural networks possess the capacity for systematic generalization, with recent studies suggesting that meta-learning approaches designed for compositionality can significantly enhance this ability. However, these insights have largely been confined to linguistic problems, leaving their applicability to other tasks an open question. In this study, we extend meta-learning for compositionality to the domain of abstract spatial reasoning. To this end, we introduce *Compositional-ARC*—a dataset designed to evaluate the capacity of models to systematically generalize from known geometric transformations (e.g., translation, rotation) of abstract two-dimensional objects to novel combinations of these transformations (e.g., translation+rotation). Our results show that a small transformer-based encoder-decoder model, trained via meta-learning for compositionality, can systematically generalize to previously unseen transformation compositions. Notably, despite having only 5.7M parameters, this model significantly outperforms state-of-the-art LLMs—including o3-mini, GPT-4o, and Gemini 2.0 Flash, which fail to exhibit similar systematic behavior—and performs on par with the winning model of the ARC prize 2024, an 8B-parameter LLM trained via test-time training. Our findings highlight the effectiveness of meta-learning in promoting systematicity beyond linguistic tasks, suggesting a promising direction toward more robust and generalizable models.

## 1 Introduction

A fundamental aspect of human cognition is the ability to *systematically generalize* from known components to novel combinations (Marcus, 2003; Lake et al., 2017). This capacity is particularly evident in language, where an infinite number of new sentences can be constructed and interpreted by extracting meaning from previously acquired expressions and rules (Chomsky, 2002; Szabó, 2012). Similarly, our spatial perception relies on systematic generalization, enabling individuals to compose learned spatial principles into novel configurations (Zhou et al., 2024; Dautriche & Chemla, 2025). For instance, once a person understands how to translate and rotate an object, they can apply these transformations in combination—translating and rotating the object simultaneously—even if they have never encountered such a composed transformation before (Fife et al., 2019).

Despite its central role in human cognition, systematic generalization remains a significant challenge in artificial intelligence (Lake & Baroni, 2018; Loula et al., 2018; Hupkes et al., 2020). While large language models have demonstrated notable progress across various domains (OpenAI, 2024; Guo et al., 2025), they often fail to combine acquired knowledge in novel scenarios, demonstrating notable difficulties with systematic generalization (Dziri et al., 2023; Ismayilzada et al., 2025; Gendron et al., 2024). The question of whether neural networks can achieve systematicity has been the subject of extensive debate (Fodor & Pylyshyn, 1988; Brakel & Frank, 2009; Calvo & Symons, 2014, *inter alia*). Recent research by Lake & Baroni (2023) demonstrates that a transformer-based encoder-

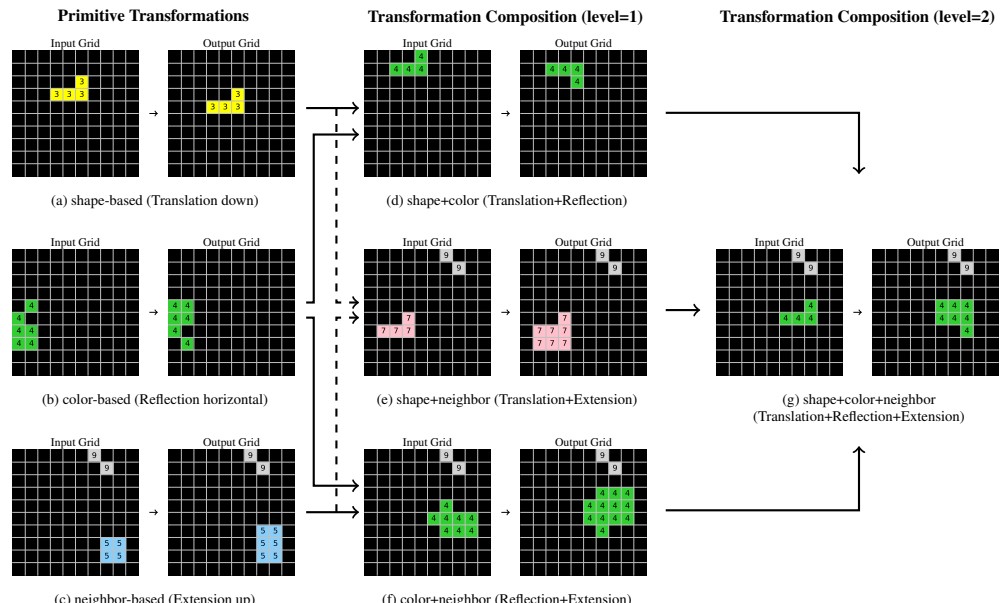

Figure 1: A conceptual overview of the data in *Compositional-ARC*. Primitive transformations refer to basic geometric transformations (e.g., translation, reflection, extension) based on an object's (a) *shape*, (b) *color*, or (c) proximity to a *neighboring* object. Pairs of these indicators, such as (d) *shape+color*, (e) *shape+neighbor*, or (f) *color+neighbor*, can be combined to form level-1 transformation compositions. Finally, all three indicators can be combined to form level-2 transformation compositions, based on the object's (g) *shape+color+neighbor*.

decoder model, trained via meta-learning for compositionality (MLC), can achieve human-like systematic generalization in processing instructions expressed in a pseudolanguage. By training the model to combine basic units of pseudolanguage into novel sequences over a stream of dynamically changing grammars, Lake & Baroni (2023) show that this model can generalize to previously unseen compositions of language (see Section 2). While this approach presents a promising direction for systematicity in neural networks, its applicability beyond linguistic contexts remains an open question.

In this study, we extend the MLC framework proposed by Lake & Baroni (2023) to the domain of abstract spatial reasoning. Inspired by the Abstraction and Reasoning Corpus (ARC) (Chollet, 2019), we introduce *Compositional-ARC*—a new dataset for assessing systematic generalization in abstract spatial reasoning. *Compositional-ARC* presents examples of basic geometric transformations (e.g., translation, rotation) applied to abstract two-dimensional objects and tests generalization to previously unseen compositions (e.g., translation+rotation; see Figure 1). Using MLC, we train a small encoder-decoder model on samples from *Compositional-ARC* and demonstrate that it can systematically generalize to unseen transformation compositions. To the best of our knowledge, this is the first application of MLC to abstract spatial reasoning. In summary, our contributions are:

1. We introduce *Compositional-ARC*—a novel dataset, inspired by ARC (Chollet, 2019), that evaluates systematic generalization in abstract spatial reasoning. The dataset includes examples of basic geometric transformations applied to abstract two-dimensional objects and tests generalization to unseen transformation compositions (see Figure 1).

2. We demonstrate that MLC enables transformer-based models to generalize to unseen compositions of geometric transformations, demonstrating its potential beyond linguistic tasks.

3. We show that a 5.7M-parameter encoder-decoder model trained via MLC significantly outperforms state-of-the-art general-purpose LLMs such as o3-mini (OpenAI, 2025), GPT-4o (Achiam et al., 2023), and Gemini 2.0 Flash (DeepMind, 2024), which fail to exhibit comparable systematic behavior on *Compositional-ARC*.

4. We find that the same MLC model performs on par with the winning model of the ARC Prize 2024, an 8B-parameter LLM trained via test-time training (Franzen et al., 2024).

## 2 BACKGROUND: META-LEARNING FOR COMPOSITIONALITY

When learning a new language, humans rely on their ability to recombine known words and expressions to interpret novel sentences (Chomsky et al., 1976; De Beule & Bergen, 2006). For instance, someone who understands the meanings of "cats drink water" and "dogs like to play" will typically also understand the meanings of "dogs drink water" and "cats like to play" (Hinzen et al., 2012). Whether language models possess a comparable degree of systematicity remains an open question, as current models, including LLMs, still struggle with tests of systematic generalization (Ismayilzada et al., 2025; Dziri et al., 2023).[1] To address these limitations, Lake & Baroni (2023) propose *meta-learning for compositionality* (MLC), a framework designed to model human-like systematic generalization in learning pseudolanguage instructions. Through a series of experiments, the authors show that models trained via MLC can achieve levels of systematicity comparable to those of humans when interpreting previously unseen pseudolanguage inputs.

**Task setup.** In their study, Lake & Baroni (2023) examine few-shot compositional tasks in which instructions, represented as sequences of pseudowords (e.g., "dax," "lug," "fep"), must be mapped to corresponding sequences of abstract symbols (see Figure 2 for an example). To understand the meaning of such instructions, an interpretation grammar needs to be deduced from a limited number of study examples. This grammar maps pseudowords to their symbolic representation through a set of compositional rewrite rules. For instance, if "dax" corresponds to a green circle, "dax fep" to three green circles, and "zup" to a red circle, then "zup fep" would denote three red circles. Importantly, the examples are designed to be highly systematic, progressing from primitive mappings to more complex compositions. The core challenge lies in the ability to generalize systematically, i.e., to reuse and combine components from the study examples (left side of Figure 2) to generate correct outputs for novel query instructions (right side of Figure 2).

**Algorithmic approach.** To achieve systematic generalization in the instruction-learning task, Lake & Baroni (2023) train a transformer-based encoder-decoder model through meta-learning for compositionality. The key idea is to train the model on a dataset of dynamically changing interpretation grammars, where the mappings from input sequences to output symbols differ across training samples. This forces the model to rely on the information conveyed in the study examples to infer the appropriate grammar of a given sample, rather than memorizing static input-output mappings across the dataset. This flexibility enables the model to adjust to novel scenarios governed by new sets of examples and rules. Moreover, the compositional structure of both study examples and queries encourages the model to internalize mechanisms for composing elements presented in the examples. After training the model over a set of 100,000 distinct interpretation grammars, it demonstrates the capacity to generalize to previously unseen instructions and grammars. For specific details regarding training procedures, we refer to Appendix C.3 and the original paper (Lake & Baroni, 2023).

While Lake & Baroni (2023) also evaluate MLC on COGS (Kim & Linzen, 2020) and SCAN (Lake & Baroni, 2018), which test systematic lexical generalization to novel word combinations, their

---

[1]For an extended literature review on systematic generalization in LLMs, please refer to Appendix A.

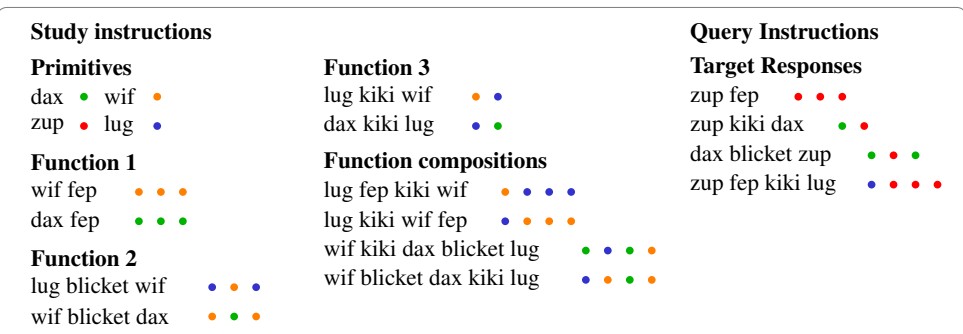

Figure 2: An example of the few-shot instruction learning task adapted from Lake & Baroni (2023). Study instructions illustrate the mapping of pseudolanguage expressions to abstract symbols.

experiments are confined to the linguistic domain. In the following section, we propose *Compositional-ARC* to show how MLC can be extended to support systematic generalization in abstract spatial reasoning, demonstrating its potential beyond linguistic tasks.

# 3 METHOD

## 3.1 COMPOSITIONAL-ARC

To test systematicity in abstract spatial reasoning, we leverage the closure property of combined geometric transformations, where the composition of two valid transformations—such as translation, rotation, and reflection—yields another valid geometric transformation (Brannan et al., 2011). Drawing inspiration from the Abstraction and Reasoning Corpus (ARC) (Chollet, 2019), we design a task in which abstract objects, defined in a two-dimensional grid environment, are subjected to basic geometric transformations and their compositions (see Figure 1 for examples). We use fixed-size $10 \times 10$ grids, each of which can be represented as a two-dimensional array of integers, where different values correspond to distinct colors. We use integers from 0 to 9, with 0 denoting a black background and the remaining integers mapping to unique colors (see Appendix B.1 for more details). Objects are defined based on color connectivity; that is, each object comprises a group of connected cells sharing the same color. Connectivity is determined by the Moore neighborhood (Bays, 2010), meaning that cells are considered connected if they are directly or diagonally adjacent. Each grid contains either one or two objects. A transformation is represented as a pair of grids, with the input grid displaying the objects before, and the output grid showing them after the geometric transformation. Each transformation affects only one of the objects in the grid. For example, in Figure 1a, a single L-shaped yellow object is translated one step downward. In Figure 1c, a square blue object in the bottom-right expands toward the neighboring top row. Objects never occlude one another nor extend beyond the boundaries of the $10 \times 10$ grids.

We limit our dataset to five basic geometric transformations and their compositions: i) translations, ii) rotations, iii) reflections, iv) extensions, and v) color changes. For our experiments, we constrain the configurations of these transformations to establish a controlled setup. Translations are limited to movements of one cell to the right or one cell downward. Rotations are restricted to 90 degrees clockwise or counterclockwise around the top-left corner of the object. We consider horizontal and vertical reflections across the object's central axis. Extensions mean that the object grows in a certain direction, and are limited to neighboring cells either leftward or upward. Color changes are restricted to changing the object's color to either red or orange. Experiments under more relaxed conditions are presented in Section 5.3. Detailed definitions of each transformation can be found in Appendix B.2.

To signal which objects undergo which transformations, we consider three types of indicators: i) *shape-based* transformations, which affect objects of a particular shape; ii) *color-based* transformations, which affect all objects of a specific color; and iii) *neighbor-based* transformations, where objects are transformed when a second, indicator object is present. For instance, in Figure 1, all L-shaped objects (similar to the object in Figure 1a) undergo a one-step downward translation. All green objects undergo a horizontal reflection, and any object sharing a grid with the gray diagonal object (e.g., as seen in Figure 1c) expands into the neighboring top row. This indicator-based approach enables the definition of transformation compositions. For example, objects that are *both* L-shaped and green undergo a one-step downward translation together with a horizontal reflection (see Figure 1d for an example). We also define different levels of composition: *level 1* combines two indicators (e.g., when an object matches the indicated shape and color, but lacks a the proximity to a neighboring object, as illustrated in Figure 1d), while *level 2* combines all three indicators, specifying the object's shape, color, and proximity to an indicator object (see Figure 1g).

To test systematicity, we present few-shot examples of primitive transformations and their *level-1* compositions, and evaluate models on previously unseen *level-2* compositions. For instance, in Figure 3, models are asked to infer the correct transformation for a previously unseen *level-2* composition of indicators, given a set of 12 *study examples* illustrating primitive transformations and their *level-1* compositions. Conceptually, our setup is similar to the few-shot compositional task introduced by Lake & Baroni (2023) (see Section 2), but it replaces the lexical interpretation grammar with a *visual* interpretation grammar. Specifically, models need to infer which indicator maps to which transformation, and how to compose them to deduce the correct final transformation. For a detailed description of how we algorithmically generate dataset samples, please refer to Appendix B.3.

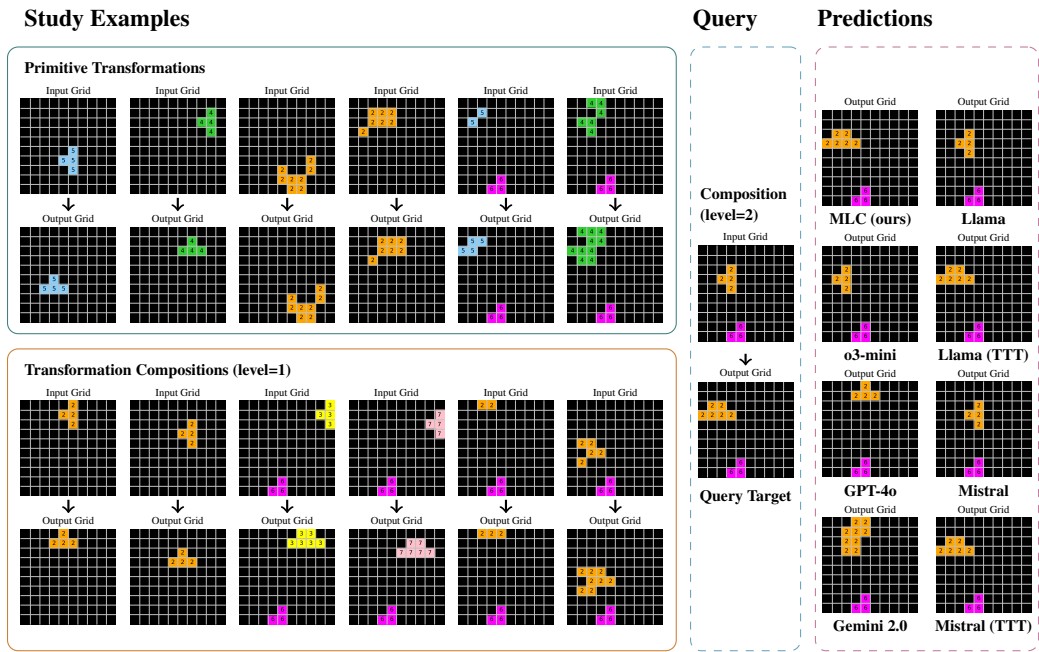

Figure 3: An episode from *Compositional-ARC*. Given a set of study examples with primitive transformations and level-1 transformation compositions, models must predict the output grid for an unseen level-2 transformation composition. Visual grammar: shape → clockwise rotation, color → translation to right, neighbor → leftward extension. Model predictions are presented to the right.

## 3.2 META-LEARNING FOR COMPOSITIONALITY IN ABSTRACT SPATIAL REASONING

To systematically generalize from known geometric transformations to previously unseen transformation compositions, we extend the meta-learning for compositionality (Lake & Baroni, 2023) framework described in Section 2. As in the original MLC approach, we train a transformer-based encoder-decoder model on a dataset of *dynamically changing* interpretation grammars. However, instead of mapping pseudolinguistic instructions to sequences of abstract symbols, we consider a *visual* interpretation grammar that associates visual indicators (object shape, color, or proximity to an indicator object) with specific geometric transformations, as described in Section 3.1. An episode in *Compositional-ARC* is defined as a set of study examples that illustrate the underlying grammar, along with query inputs for which the correct outputs must be inferred. For instance, the episode in Figure 3 contains 12 study examples: six *primitive* transformations (two per indicator type) and six *level-1* compositions (two per composition type). Given the study examples, the model is asked to predict output grids for previously unseen *level-2* compositions. By training over a series of episodes with *changing* visual interpretation grammars, the model needs to abstract and recombine information from the examples in order to predict the correct query transformation composition, as it cannot rely on fixed mappings from indicators to transformations.

**Encoding and positional embedding.** Each episode is presented to the model as a sequence of input-output grid pairs (study examples), followed by a query input grid, for which the model must generate the corresponding output grid (see Figure 3). To encode the two-dimensional grids, we divide each $10 \times 10$ grid into $2 \times 2$ patches (left to right, top to bottom), yielding 25 patches per grid (Dosovitskiy et al., 2021). Each patch is mapped to a unique embedding vector. Since each grid cell can take integer values from 0 to 9, a $2 \times 2$ patch can yield up to 10,000 distinct configurations, resulting in 10,000 possible embedding vectors. Two special tokens, | and →, are introduced to mark the boundaries between study examples and the input-output grids, respectively. The decoder vocabulary comprises two additional tokens for the start and end of a sequence (SOS and EOS). To encode positional information, we use standard learnable 1D positional embeddings that capture the order of grid pairs, as well as a second set of learnable 2D positional embeddings applied to grid

patches. These 2D embeddings are decomposed into separate row and column components, which are added to each patch embedding to capture two-dimensional spatial information.

**Training procedure.** The model is trained on a large set of episodes, each defined by a unique *visual* interpretation grammar. In each episode, the model is provided with a sequence of study examples and tasked with predicting the output grid for a given input query (see Figure 3). Following Lake & Baroni (2023), we include an auxiliary copy task during training, in which the model must also reproduce the output grids of each study example. We employ a model with three layers each in the encoder and decoder, eight attention heads per layer, input and hidden embeddings of size 128, a feedforward hidden size of 768, and GELU (Hendrycks & Gimpel, 2016) activations. In total, the model has 5.7 million trainable parameters. To promote robustness in the decoder, we introduce minor perturbations by randomly altering the color of individual cells in the target output query with a small probability (0.001). Unlike Lake & Baroni (2023), we do not incorporate systematic noise to model inductive biases observed in human learning. Further implementation details regarding the training procedure and hyperparameters can be found in Appendix C.

## 4 EXPERIMENTAL SETUP

### 4.1 TASK SETUP

We consider two task setups in this work. The first, denoted as "*3-Shot*," is a standard few-shot learning task where models must generate an output grid for a query input that performs a *level-2* transformation composition. This prediction is based on three examples illustrating the same *level-2* transformation. A visual representation of this setup is provided in Figure 5 in the Appendix. This task evaluates the model's ability to infer geometric transformations from a limited set of examples.

The second setup, denoted as "*Systematicity*," focuses on compositional generalization and differs from the first in the type of few-shot examples presented. As mentioned in Section 3.1, the idea is to test whether models can infer novel compositions from known geometric transformations. To this end, we replace the *level-2* few-shot examples with a set of *primitive* transformations plus *level-1* transformation compositions, and query the model to predict the previously unseen *level-2* transformation composition, as illustrated in Figure 3. Specifically, we present six *primitive* transformations—two examples for each indicator (*shape-based*, *color-based*, *neighbor-based*)—and six *level-1* transformation compositions, two examples for each *level-1* indicator composition (*shape+color*, *shape+neighbor*, *color+neighbor*).

We generate 100,000 episodes, each comprising three few-shot examples for the "*3-Shot*" task, 12 systematic study examples for the "*Systematicity*" setup, and ten query input-output grid pairs demonstrating the final *level-2* transformation composition. Each episode is characterized by a *unique* visual interpretation grammar. For instance, in one episode, yellow objects are translated downward by a single cell, while in another, yellow objects are reflected horizontally. To train our encoder-decoder model via MLC, we split the data into 82,908 training, 8,546 validation, and 8,546 test episodes. Importantly, the data splits are constructed such that the geometric transformations involved in the final query *level-2* compositions **differ** between the training and evaluation sets. For instance, while the model is trained on basic transformations and a series of transformation compositions (e.g., *translation+rotation+reflection*), it is tested out-of-distribution on compositions **not** seen during training (e.g., *translation+rotation+extension*). For comprehensive statistics on the dataset splits, please refer to Appendix B.4, especially Table 7.

### 4.2 LARGE LANGUAGE MODELS

**General-purpose LLMs.** In addition to the model trained via MLC, we evaluate three state-of-the-art general-purpose LLMs on the test set of our proposed dataset: o3-mini (low) (OpenAI, 2025), GPT-4o (Achiam et al., 2023), and Gemini 2.0 Flash (DeepMind, 2024). To textually prompt the models for a given episode, we represent grids as two-dimensional arrays, consistent with prior work (Moskvichev et al., 2023). We also test a multimodal setup in which both an image of the study examples and the input query are provided alongside the text prompt. Due to financial constraints, each model is evaluated on a single test query for each of the 8,546 episodes in the test set. All textual and visual prompts, specific model versions, and decoding parameters are detailed in Appendix D.2.

Table 1: Comparison of model performance across the two different task setups. We report exact match accuracy, color accuracy, and shape accuracy as described in Section 4.3).

| | Model | Exact Match Accuracy [%] | Color Accuracy [%] | Shape Accuracy [%] |
|---|---|---|---|---|
| **3-Shot** | GPT-4o | 22.28 | 99.67 | 57.02 |
| | + *image* | 19.42 | 99.75 | 54.56 |
| | Gemini 2.0 Flash | 30.08 | 99.92 | 52.34 |
| | + *image* | 17.19 | 99.79 | 35.86 |
| | o3-mini (low) | 64.04 | 99.89 | 68.74 |
| | Llama-3.2-3B-ReARC | 85.85 | 98.57 | 86.05 |
| | Mistral-NeMO-Minitron-8B-Full | 95.71 | 99.85 | 96.78 |
| | MLC (ours) | **99.92** | **100.00** | **99.92** |
| **Systematicity** | GPT-4o | 0.99 | 99.23 | 9.82 |
| | + *image* | 0.86 | 97.94 | 7.50 |
| | Gemini 2.0 Flash | 2.66 | 99.68 | 12.81 |
| | + *image* | 2.05 | 99.28 | 9.60 |
| | o3-mini (low) | 0.53 | 99.10 | 5.65 |
| | Llama-3.2-3B-ReARC | 0.87 | 99.94 | 2.54 |
| | + *test-time training* | 73.70 | **100.00** | 86.88 |
| | Mistral-NeMO-Minitron-8B-Full | 0.70 | 99.99 | 9.75 |
| | + *test-time training* | 78.20 | **100.00** | **88.26** |
| | MLC (ours) | **78.26** | 97.88 | 80.49 |

**Domain-specific LLMs.** We further consider two LLMs specifically tailored to ARC-style data: (i) Llama-3.2-3B-ReARC, fine-tuned on the re-ARC dataset (Hodel, 2024)—an extension of 1,000 additional generated examples per sample in ARC—and (ii) Mistral-NeMO-Minitron-8B-Full, trained on a broad range of ARC-style data, including re-ARC, Concept-ARC (Moskvichev et al., 2023), and ARC-Heavy (Li et al., 2025). These models were proposed by Franzen et al. (2024) and placed 1st in the ARC prize 2024.[2] Note that in addition to fine-tuning, these models use an ARC-customized tokenizer, extensive data augmentation during training and inference, a generation procedure that leverages depth-first search to produce multiple solution candidates, and a refined candidate-selection step. The authors also employ test-time training (TTT), which further fine-tunes models on the few-shot input–output grid pairs from the final test set. We use both models with their default parameters. For additional details, please refer to the original paper (Franzen et al., 2024) or Appendix D.2.

## 4.3 EVALUATION METRICS

To evaluate the quality of the generated output grids, we use three different metrics: i) exact match accuracy, ii) color accuracy, and iii) shape accuracy. Exact match accuracy requires that a prediction is accurate only if every cell matches the target grid. Color accuracy checks whether predicted objects match target colors, ignoring shape and location. Shape accuracy checks whether predicted objects match target shapes, ignoring color and location. Formal definitions are provided in Appendix D.1.

## 5 RESULTS

In Table 1, we report the performance of the model trained via MLC, alongside the LLMs we evaluate on the two task setups, as described in Section 4.1.

**Standard few-shot learning task.** We begin by examining model performance on the "*3-Shot*" task, where models are given three input-output examples illustrating the final transformation composition (see Figure 5 in the Appendix). Despite this guidance and the relatively simple transformations involved, general-purpose LLMs such as GPT-4o and Gemini 2.0 Flash struggle with the task: GPT-4o reaches an accuracy of only 22.28%, while Gemini 2.0 Flash performs slightly better at 30.08%. The long-chain-of-thought model o3-mini achieves a modest accuracy of 64.04%. In

---

[2]https://arcprize.org/competitions/2024

contrast, domain-specific models such as Llama-3.2-3B-ReARC, and Mistral-NeMO-Minitron-8B-Full perform significantly better. While Llama-3.2-3B-ReARC achieves an accuracy of 85.85%, Mistral-NeMO-Minitron-8B-Full reaches up to 95.71%. Note that we do not employ test-time training in this setup, as it would contradict the out-of-distribution test setup described in Section 4.1. Notably, the 5.7M-parameter encoder-decoder model trained via MLC outperforms both general-purpose and domain-specific LLMs, with an accuracy of 99.92%, despite having only a fraction of the parameters. We further find that all models predict object color nearly perfectly. For GPT-4o and Gemini 2.0 Flash, we observe that shape accuracy is significantly higher than exact match accuracy. This discrepancy suggests that while these models are often able to predict the correct shape and color of an object, they frequently fail to accurately predict its final position. Interestingly, both models show lower accuracy when visual input is added to the textual prompt, likely due to modality alignment challenges (Masry et al., 2025) or limitations in leveraging the visual content for reasoning.

**Systematicity task.** In the "*Systematicity*" task, models are asked to infer the correct final transformation composition from a set of study examples that represent more basic, decomposed transformations (see Figure 3 for an example). As shown in Table 1, all general-purpose LLMs perform poorly on this task. For instance, GPT-4o achieves an accuracy of 0.99%, while Gemini 2.0 Flash reaches 2.66%. Interestingly, o3-mini, the best-performing general-purpose model on the "*3-Shot*" task, performs worst in this setting, with an accuracy of only 0.53%. For the domain-specific LLMs, we find that test-time training (TTT)—where models are additionally fine-tuned on the study examples' input-output grid pairs of the test set—significantly improves performance. While Llama-3.2-3B-ReARC achieves only 0.70% accuracy without TTT, performance increases to 73.70% with TTT. Similarly, Mistral-NeMO-Minitron-8B-Full's accuracy increases from 0.70% to 78.20% with TTT. We hypothesize that training on the systematic study examples of the test data (demonstrating *primitive* and *level-1* transformations) teaches the models how to abstract and compose transformations for the final input query. We further find that the much smaller 5.7M-parameter MLC model performs on par with the domain-specific LLMs trained via TTT, slightly outperforming Mistral-NeMO-Minitron-8B-Full with an accuracy of 78.26%. Importantly, as described in Section 4.1, the MLC model has never seen the specific *level-2* compositions of the test data during training, but was instead optimized on a distinct set of transformation compositions (see data split for seed 1860; Table 7 in the Appendix). Consistent with our findings from the 3-shot learning task, models generally succeed in predicting the correct object colors. However, shape accuracy declines markedly. A qualitative example of the models' predictions is shown in Figure 3, with additional examples in Figures 8–9 in the Appendix. The strong performance of the small MLC model highlights the effectiveness of this training strategy in promoting systematic generalization to novel transformation compositions. The model not only learns to infer a visual interpretation grammar from a limited number of study examples but also generalizes to novel transformation compositions that it has never encountered during training.

## 5.1 CONSISTENCY ACROSS DATA SPLITS

To ensure that the strong performance of MLC, as reported in Table 1, is not the result of a favorable data split, we train and evaluate the model on three additional, independently generated data splits for each task configuration—resulting in four distinct models per task setup. Detailed descriptions of these data splits are provided in Table 7 in the Appendix. Table 2 summarizes the average accuracy and corresponding standard deviation across all four splits. For the standard three-shot learning task,

Table 2: Average accuracy and standard deviation across the four different data splits. For the systematicity task, we ablate different components of the training procedure to assess their individual contributions and overall impact.

| Model | Exact Match Accuracy [%] | Color Accuracy [%] | Shape Accuracy [%] |
|---|---|---|---|
| MLC (3-Shot) | 98.78 ± 1.99 | 100.00 ± 0.00 | 98.79 ± 1.98 |
| MLC (Systematicity) | **86.73 ± 6.03** | 99.36 ± 0.70 | **87.55 ± 5.45** |
|   - no copy task | 69.05 ± 9.23 | 99.43 ± 0.38 | 70.60 ± 9.23 |
|   - no primitives | 75.27 ± 12.95 | **99.56 ± 0.50** | 76.92 ± 11.23 |
|   - no level-1 compositions | 21.01 ± 19.07 | 94.72 ± 7.41 | 23.03 ± 19.08 |

MLC consistently achieves high accuracy, with a mean of 98.78% and a standard deviation of 1.99%. Similarly, for the systematicity task, the model demonstrates robust generalization, achieving an even higher average accuracy than on the initial data split, with a mean of 86.73%.

**Ablation studies.** To gain deeper insights into the factors influencing model performance, we conduct a series of ablation studies. First, we evaluate the impact of removing the auxiliary copy task from the training objective—a setup in which the model is trained not only to predict the output grid for a given input query but also to reproduce the output grid of each study example (refer to Section 3.2). Removing this auxiliary task results in a notable decrease in accuracy from 86.73% to 69.05%. This decline underscores the importance of the copy task in promoting systematic generalization, aligning with the findings of Lake & Baroni (2023). Subsequently, we assess the role of study examples in model performance. Removing *primitive* transformations from the study examples (see Figure 3) results in a moderate reduction in performance, with an average accuracy of 75.27%. This suggests that examples involving only *level-1* transformation compositions are, to some extent, sufficient for allowing the model to generalize to more complex *level-2* compositions. However, removing *level-1* transformation compositions leads to a severe performance degradation, reducing accuracy to 21.01%. We hypothesize that this is due to the increased complexity of composing three primitive operations directly into a *level-2* transformation, as opposed to building on intermediate *level-1* compositions.

## 5.2 ERROR ANALYSIS

To characterize model behavior on the systematicity task, we analyze the models' prediction errors. Figure 4 shows the relative frequency of common error types across models. We consider the following error categories: (i) *Format* errors (output not a valid $10 \times 10$ grid with cell values in $0, \ldots, 9$); (ii) *No Transformation* (output identical to input); (iii) *Primitive* (a primitive transformation is applied instead of the target level-2 composition); (iv) *Level-1* (a level-1 composition is applied instead of the level-2 composition); (v) *Invalid Position* (correct color and shape, wrong position); (vi) *Invalid Shape* (correct color, incorrect shape); and (vii) *Other* (e.g., wrong number of objects, or objects with both incorrect shape and color). Models show distinct error profiles. General-purpose LLMs (GPT-4o, Gemini 2.0 Flash, o3-mini) most often predict incorrect shapes that do not match any primitive or level-1 outcome; for o3-mini, over 30% of errors involve applying a primitive instead of a level-2 composition, and with image input more than 20% of GPT-4o's errors are format violations. Llama-3.2-3B-ReARC mainly copies the input (no transformation), whereas Mistral-NeMO-Minitron-8B-Full most often applies a primitive instead of the target level-2 composition. After test-time training on the study examples (Section 4.2), errors of both domain-specific LLMs most often involve level-1 predictions. The MLC model rarely produces primitive or level-1 outputs; instead, it fails mainly by predicting an incorrect shape. Exact percentages by model and error type, and a breakdown of primitive and level-1 errors, are reported in Tables 5 and 6 in the Appendix.

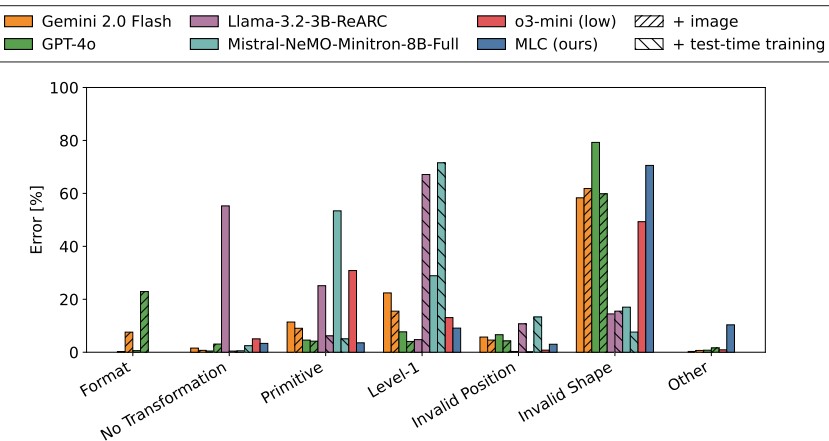

Figure 4: Error distribution by error category across models. Bars show the fraction of prediction errors assigned to each error category.

## 5.3 INCREASING DATASET COMPLEXITY

So far, *Compositional-ARC* has been restricted to i) translations of one cell to the right or downward; ii) 90-degree clockwise or counterclockwise rotations; iii) horizontal and vertical reflections; iv) extensions to neighboring cells leftward or upward; and v) color changes to red or orange. We analyze whether the MLC model still systematically generalizes when we increase the variety of transformations, and therefore the dataset complexity. To this end, we introduce a new dataset that additionally allows translations of one or two cells in any direction (left, right, upward, downward), extensions to neighboring cells in any direction, and color changes to red, orange, yellow, and green. We generate 100,000 episodes and split the data as described in Section 4.1; exact dataset statistics are given in Table 8 in the Appendix. We then train an MLC model following the procedure in Section 3.2. Even on this more diverse dataset, the MLC model systematically generalizes to unseen transformation compositions, achieving an exact match accuracy of $88.10\%$ on the test set, a color accuracy of $99.83\%$, and a shape accuracy of $88.25\%$.

## 6 RELATED WORK

**Meta-learning.**    Meta-learning aims to improve a model's ability to adapt to novel tasks by leveraging experience over multiple training episodes (Thrun & Pratt, 1998; Hospedales et al., 2022). It has been successfully applied to various tasks, such as few-shot learning (Mishra et al., 2018), continual learning (Javed & White, 2019; Lee et al., 2023; Irie et al., 2025), and reinforcement learning (Duan et al., 2016; Wang et al., 2017; Mishra et al., 2018). Related to our work, meta-learning has been used to improve systematic generalization. Lake & Baroni (2018) showed that traditional sequence-to-sequence models struggle with compositional skills, but incorporating meta-learning can significantly improve performance (Lake, 2019; Conklin et al., 2021). Recent work argues that giving models the opportunity to practice skills via meta-learning is crucial for addressing challenges such as systematic generalization, among others (Irie et al., 2025). Our method builds on meta-learning strategies inspired by Lake & Baroni (2023), extending them to the domain of abstract spatial reasoning.

**ARC-like puzzles.**    The abstraction and reasoning corpus (ARC) (Chollet, 2019) is a benchmark designed to evaluate a model's capacity to generalize to novel scenarios with limited to no prior knowledge. Based on a set of few-shot examples, models are required to infer transformations of abstract objects or patterns within two-dimensional grids. Unlike ARC, which encompasses a broad range of complex transformations, our work deliberately narrows the scope to the five fundamental geometric transformations described in Section 3.1, focusing instead on the aspect of systematicity. Several ARC variants have been proposed, including 1D-ARC (Xu et al., 2024), Mini-ARC (Kim et al., 2022), ConceptARC (Moskvichev et al., 2023) and MC-LARC (Shin et al., 2024). However, to the best of our knowledge, *Compositional-ARC* is the first to focus on compositional generalization.

## 7 CONCLUSION

In this work, we extend the meta-learning for compositionality framework proposed by Lake & Baroni (2023) to the domain of abstract spatial reasoning. To this end, we introduce *Compositional-ARC*—a novel dataset designed to evaluate systematicity in this field. Our experiments demonstrate that models trained via MLC can systematically generalize to novel compositions of geometric transformations. Moreover, a small MLC model outperforms state-of-the-art general-purpose LLMs on *Compositional-ARC*, and performs on par with domain-specific LLMs trained via test-time training. Our findings suggest that MLC presents a promising direction for enabling systematic generalization in language models across diverse domains.

**Limitations & Future directions.**    A notable limitation of the current version of *Compositional-ARC* is its restriction to fixed-size grids and limited number of transformations. While it is possible to extend the dataset to more diverse grid setups, it is currently unclear how MLC would perform on more complex transformations. A promising direction for future work is to train an additional model that learns how to decompose complex ARC-like problems into primitive transformations, and then train MLC on these primitives to generalize to unseen, more complex transformation compositions.

## REPRODUCIBILITY STATEMENT

To ensure the reproducibility of our work, we make all code publicly available at: https://github.com/mainlp/C-ARC. This enables users to reproduce the data described in Section 3.1 and train models via MLC for the task, as outlined in Section 3.2. Details about the training procedures and hyperparameters are provided in Section 3.2 and Appendix C. Specifics on prompts, model versions, and decoding parameters are given in Appendix D.2. Our datasets are publicly available at: https://huggingface.co/datasets/mainlp/Compositional-ARC, with further details presented in Section 3.1, Section 4.1, and Appendix B. Finally, Appendix C.2 outlines the software and computational resources used for model training.

## ACKNOWLEDGMENTS

We express our gratitude to the members of the MaiNLP lab for their invaluable feedback. Furthermore, we thank the anonymous reviewers for their insightful comments and suggestions. We gratefully acknowledge that experiments involving API calls to GPT-4o and o3-mini were supported by a compute grant from OpenAI. The authors also acknowledge the scientific support and HPC resources provided by the Erlangen National High Performance Computing Center (NHR@FAU) of the Friedrich-Alexander-Universität Erlangen-Nürnberg (FAU) under the NHR project b217dd. NHR funding is provided by federal and Bavarian state authorities. NHR@FAU hardware is partially funded by the German Research Foundation (DFG) – 440719683. Finally, we acknowledge the support for BP through the ERC Consolidator Grant DIALECT 101043235.

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

## A    SYSTEMATIC GENERALIZATION IN LLMS

The question of whether neural networks, and more recently large language models, have the capacity to generalize systematically from known components to novel combinations has been, and continues to be, the subject of extensive debate (Fodor & Pylyshyn, 1988; Brakel & Frank, 2009; Lake & Baroni, 2023; Mannekote, 2024, *inter alia*). This section offers an extended literature review on systematic generalization in LLMs, presenting an overview of recent studies that assess systematicity in current language models.

Following Hupkes et al. (2020), different aspects of compositionality need to be distinguished. *Systematicity* refers to the capacity to recombine known parts and rules into novel combinations (Szabó, 2012; Hupkes et al., 2020; Lake & Baroni, 2023). More formally, this capacity can be defined as:

> **Definition 1** (Systematic generalization). The capacity to recombine previously observed or learned parts and rules, i.e., primitives $e_1, e_2, \ldots, e_n$, to generalize to novel, previously unseen compositions of them (e.g., $e_1 \times e_2$).

According to Hupkes et al. (2020), systematicity is different from other aspects of compositionality, such as *productivity*: the capacity to predict expressions beyond the length of those already encountered, or *substitutivity*: the ability to handle synonym substitutions. For more details on the different aspects of compositionality, we refer to the original study by Hupkes et al. (2020).

**Systematicity in current LLMs.**    A growing body of research evaluates whether large language models satisfy the criteria of systematicity, i.e., whether they can generalize systematically from known or previously seen components to novel combinations. Thomm et al. (2024) study whether LLMs such as LLaMA (Touvron et al., 2023), GPT-4 (Achiam et al., 2023), and Gemini-Pro (Team et al., 2023) can solve compositional algorithmic tasks by reusing previously encountered primitives. Training a small LLaMA-style model from scratch on four compositional algorithmic tasks shows that, while the model is able to learn all sub-tasks or primitives reliably, it fails to properly compose them. Instead, the model exhibits extreme sample inefficiency: it is able to solve the compositional task only when the amount of training data is increased by almost one order of magnitude. The authors further present a complexity-theoretic argument that gradient-descent training of fixed-depth feedforward models is asymptotically data-inefficient on combinatorial problems. Prompt-based evaluations of GPT-4 and Gemini-Pro further show that these models struggle with the tasks, even when strong hints are provided or techniques such as chain-of-thought (CoT) prompting (Wei et al., 2022) are used. Dziri et al. (2023) also highlight compositional limits of LLMs on multiplication, logic-grid puzzles, and a dynamic-programming task: performance is near-perfect in-distribution but collapses as computation graphs deepen or branch beyond training complexity. Petruzzellis et al. (2024) complement these findings with a systematic study of LLMs' performance on algorithmic tasks where both the number of operands per operation and their nesting depth can be controlled. While the authors show that LLMs such as GPT-4 fail on highly nested, multi-operand formulas, they find that advanced prompting strategies such as zero-shot CoT (Kojima et al., 2022) with self-consistency (Wang et al., 2023a) can improve performance on less complex compositions.

Zhao et al. (2024) introduce MATHTRAP, where standard problems from GSM8K (Cobbe et al., 2021) and MATH (Hendrycks et al., 2021) are modified with logical "traps" (e.g., undefined concepts, missing conditions, contradictions) that require combining ordinary math-solving competence with the capacity to identify such inconsistencies. The authors show that LLMs such as Llama-3 (70B) (Grattafiori et al., 2024), Claude3-Opus (Anthropic, 2024), and GPT-4 score well on the original

problems and on standalone questions about the logical inconsistencies presented, yet their accuracy drops dramatically on such trap problems. Prompting that warns about traps, few-shot demonstrations, fine-tuning, and OpenAI's o1 "slow thinking" (Jaech et al., 2024) improve performance but still leave a substantial gap to human performance on the task.

In semantic parsing, where natural language must be translated into a structured (often symbolic) form, systematic generalization plays an important role when novel compositional queries that have not been seen during training are introduced (Mannekote, 2024). Schmidt et al. (2025) propose CompoST, a controlled semantic-parsing benchmark for evaluating systematic generalization in question answering over linked data from DBpedia (Mendes et al., 2012). Models need to map natural-language questions to SPARQL queries where all atomic graph-pattern constituents have been presented, while novel combinations appear at test time. Across zero-shot, few-shot, and fine-tuned settings on three difficulty splits, performance drops sharply as structural deviation increases. The authors conclude that current LLMs struggle to systematically recombine known SPARQL constituents into correct queries, indicating weak systematic generalization in this domain. In contrast, Drozdov et al. (2023) show that models can achieve high systematic generalization on semantic parsing datasets such as CFQ (Keysers et al., 2020) and COGS (Kim & Linzen, 2020) when explicitly prompted to decompose problems. The authors introduce dynamic least-to-most prompting, where models first decompose the input and then solve subproblems sequentially. Compared to standard few-shot prompting, least-to-most prompting achieves near-SOTA OOD performance on CFQ and COGS, suggesting that systematicity is not reliably expressed by default but can be elicited. Yang et al. (2024) study "order-$n$" compositional instructions derived via self-instruct (Wang et al., 2023b). Training on higher-order compositions improves performance on lower-order ones, but training on simpler orders does not transfer to longer compositions, revealing an asymmetry typical of non-systematic learners. Sakai et al. (2025) introduce Ordered CommonGen, where four known concepts must be embedded in a sentence in a specified order across permutations. While unordered concept coverage is high, ordered coverage remains substantially lower even for LLMs, indicating difficulty in faithfully recombining familiar concepts under novel structural constraints. Ismayilzada et al. (2025) extend systematicity tests to morphology in agglutinative languages (Turkish, Finnish): LLMs struggle to generate or validate novel morpheme compositions, particularly for nonce roots and longer affix chains, and performance degrades with compositional length.

Two data-centric accounts help explain when LLMs succeed or fail. Wold et al. (2025) argue that systematic generalization scales with the *information entropy* of the training distribution over primitives; in modified SCAN, higher-entropy coverage of verbs and contexts yields smooth improvements in systematic generalization. Chang et al. (2025) formalize the *coverage principle*, showing that systematicity in transformer-based models largely reduces to substituting functionally equivalent fragments observed in shared contexts. They show that data requirements for multi-hop systematicity grow at least quadratically in component set size and are largely insensitive to parameter scaling.

**Summary.** Overall, the surveyed studies suggest that current LLMs do not reliably exhibit human-like systematic generalization under standard training and evaluation: performance often correlates with training data coverage and degrades on novel compositions. However, prompting techniques for explicit decomposition and compositional training curricula might be able to elicit systematicity, consistent with a view that compositional abilities are partly latent but not automatically deployed.

## B DATASET

In this work, we present *Compositional-ARC*, a dataset designed to study systematicity in abstract spatial reasoning. As outlined in Section 3.1, *Compositional-ARC* evaluates a model's capacity to systematically generalize learned geometric transformations (e.g., translation, rotation) of two-dimensional objects to novel compositions of these transformations (e.g., translation+rotation). The subsequent sections offer a detailed description of the dataset, including formal definitions of the grid-based environment and the set of transformations it includes.

### B.1 GRID SETUP

We define the structure of the $10 \times 10$ grid environment and the notion of objects within it. Each grid is represented as a matrix $\boldsymbol{X} \in \mathbb{N}^{10 \times 10}$, where each element corresponds to a cell with a discrete

color value. Objects are defined based on color connectivity using the Moore neighborhood (Bays, 2010).

---

**Definition 2** (Grid & Object). Let $\boldsymbol{X} \in \mathbb{N}^{10 \times 10}$ be a matrix with rows $i$ and columns $j$, referred to as a *grid*, where each element $\boldsymbol{X}_{ij} \in \{0, \ldots, 9\}$. The value $\boldsymbol{X}_{ij} = 0$ represents a background cell, and values $\boldsymbol{X}_{ij} \in \{1, \ldots, 9\}$ represent object colors.

An *object* is a set of coordinates
$$\mathbb{O} \subseteq \{0, \ldots, 9\}^2$$
such that each $(i, j) \in \mathbb{O}$ satisfies $\boldsymbol{X}_{ij} = c$, and the elements in $\mathbb{O}$ form a single connected component.

Two elements $\boldsymbol{X}_{ij}$ and $\boldsymbol{X}_{kl}$ are considered *connected* if:
$$\max(|i - k|, \ |j - l|) \leq 1$$

---

We define the following color mapping: $0 \to$ black, $1 \to$ red, $2 \to$ orange, $3 \to$ yellow, $4 \to$ green, $5 \to$ blue, $6 \to$ purple, $7 \to$ pink, $8 \to$ cyan, and $9 \to$ gray.

## B.2 GEOMETRIC TRANSFORMATIONS

We formally define the five basic geometric transformations used in our dataset: translation, rotation, reflection, extension, and color change. Each transformation operates on objects within the grid environment as defined in Appendix B.1. A transformation is considered *valid* if all transformed coordinates lie within the grid bounds and do not overlap with existing objects in the original grid.

**Translation.** Moves an object by one cell along a specified direction (downward or rightward). A formal definition is given in the text box below.

---

**Definition 3** (Translation). Let $\mathbb{O} \subseteq \{0, \ldots, 9\}^2$ be an object in a grid $\boldsymbol{X} \in \mathbb{N}^{10 \times 10}$, and let $\boldsymbol{v} = (\boldsymbol{v}_1, \boldsymbol{v}_2) \in \{(1, 0), (0, 1)\}$ be the translation direction (downward or rightward).

The translated object is:
$$T_{\text{trans}, \boldsymbol{v}}(\mathbb{O}) = \{(i + \boldsymbol{v}_1, j + \boldsymbol{v}_2) \mid (i, j) \in \mathbb{O}\}$$

The translation is *valid* if:
$$\forall (i', j') \in T_{\text{trans}, \boldsymbol{v}}(\mathbb{O}), \quad 0 \leq i', j' < 10, \quad \boldsymbol{X}_{i'j'} = 0$$

---

**Rotation.** Rotates an object $90°$ clockwise or counterclockwise around the top-left of its bounding box. A more formal definition is given in the text box below.

---

**Definition 4** (Rotation). Let $\mathbb{O} \subseteq \{0, \ldots, 9\}^2$ be a set of grid cells with row–column coordinates $(i, j)$. Let $i_0 = \min_{(i,j) \in \mathbb{O}} i$ and $j_0 = \min_{(i,j) \in \mathbb{O}} j$. We set the pivot $P = (i_0, j_0)$ as the top-left of the bounding box.

For each $(i, j) \in \mathbb{O}$, we specify the offset from the pivot as:

$$(\Delta i, \Delta j) = (i - i_{\min}, \ j - j_{\min}).$$

We define a rotation by $\pm 90°$ as:

$$R_{+90°}(\Delta i, \Delta j) = (\Delta j, \ -\Delta i), \qquad R_{-90°}(\Delta i, \Delta j) = (-\Delta j, \ \Delta i),$$

where $+90°$ is clockwise and $-90°$ is counterclockwise under the row-down convention.

Given a $90°$ rotation, either clockwise or counterclockwise, the rotated object is:

$$T_{\text{rot},\pm 90°}(\mathbb{O}) = \big\{ \, (i_{\min} + \Delta i, \ j_{\min} + \Delta j) \ \big| \ (i, j) \in \mathbb{O} \, \big\}.$$

The rotation is *valid* if:

$$\forall (i', j') \in T_{\text{rot},\theta}(\mathbb{O}), \quad 0 \le i', j' < 10, \quad x_{i'j'} = 0$$

---

**Reflection.** Reflects an object across its vertical or horizontal axis, reversing the relative positions of its coordinates while preserving overall structure.

---

**Definition 5** (Reflection). Let $\mathbb{O} \subseteq \{0, \ldots, 9\}^2$ be an object in a grid $\boldsymbol{X} \in \mathbb{N}^{10 \times 10}$, and let $d \in \{\text{horizontal}, \text{vertical}\}$ indicate the axis of reflection.

Let:

$$i_{\min} = \min\{i \mid (i,j) \in \mathbb{O}\}, \quad i_{\max} = \max\{i \mid (i,j) \in \mathbb{O}\}$$
$$j_{\min} = \min\{j \mid (i,j) \in \mathbb{O}\}, \quad j_{\max} = \max\{j \mid (i,j) \in \mathbb{O}\}$$

Then the reflected object is:

$$T_{\text{ref},d}(\mathbb{O}) = \begin{cases} \{(i_{\max} - (i - i_{\min}), \ j) \mid (i,j) \in \mathbb{O}\} & \text{if } d = \text{horizontal} \\ \{(i, \ j_{\max} - (j - j_{\min})) \mid (i,j) \in \mathbb{O}\} & \text{if } d = \text{vertical} \end{cases}$$

The reflection is *valid* if:

$$\forall (i', j') \in T_{\text{ref},d}(\mathbb{O}), \quad 0 \le i', j' < 10, \quad \boldsymbol{X}_{i'j'} = 0$$

---

**Extension.** Adds a new cell in the upward or leftward direction for each coordinate in the object.

---

**Definition 6** (Extension). Let $\mathbb{O} \subseteq \{0, \ldots, 9\}^2$ be an object in a grid $\boldsymbol{X} \in \mathbb{N}^{10 \times 10}$, with color $c > 0$. Let $d \in \{\text{up, left}\}$ indicate the extension direction.

Let the set of new cells adjacent to the object in direction $d$ be:

$$N_d(O) = \begin{cases} \{(i-1, j) \notin \mathbb{O} \mid (i,j) \in \mathbb{O}, \ i > 0, \ x_{i-1,j} = 0\} & \text{if } d = \text{up} \\ \{(i, j-1) \notin \mathbb{O} \mid (i,j) \in \mathbb{O}, \ j > 0, \ x_{i,j-1} = 0\} & \text{if } d = \text{left} \end{cases}$$

Then the extended object is:
$$T_{\text{ext},d}(\mathbb{O}) = \mathbb{O} \cup N_d(\mathbb{O})$$

The extension is *valid* if:
$$\forall (i', j') \in N_d(\mathbb{O}), \quad 0 \leq i', j' < 10, , \quad \boldsymbol{X}_{i'j'} = 0$$

All new cells $(i', j') \in N_d(\mathbb{O})$ are assigned the color of the original object:
$$\boldsymbol{X}'_{i'j'} = c$$

---

**Color change.** Alters the color of an object to either red or orange, without changing its structure or position.

---

**Definition 7** (Color Change). Let $\mathbb{O} \subseteq \{0, \ldots, 9\}^2$ be an object in a grid $\boldsymbol{X} \in \mathbb{N}^{10 \times 10}$, with color $c > 0$. Let $c' \in \{1, 2\}$ be the new color (representing red or orange).

The resulting grid $X'$ is given by:

$$\boldsymbol{X}'_{ij} = \begin{cases} c' & \text{if } (i,j) \in \mathbb{O} \\ \boldsymbol{X}_{ij} & \text{otherwise} \end{cases}$$

---

### B.3 DATASET GENERATION

To generate episodes that comprise primitive transformations, level-1 transformation compositions, and level-2 transformation compositions, we developed a script that systematically generates the corresponding input-output grid pairs for each transformation. The complete code repository for data generation is publicly available at: https://github.com/mainlp/C-ARC. In the following, we provide a brief overview of the procedure used to generate input-output grid pairs for each sample within an episode. As detailed in Section 3.1 and Appendix B.2, we consider five basic geometric transformations, along with three types of transformation indicators: shape-based, color-based, and neighbor-based. These allow us to define a total of ten distinct transformation triplets, each mapping the indicators to corresponding transformations (e.g., shape-based: translation, color-based: reflection, neighbor-based: extension). For each episode, a transformation triplet is uniformly sampled from this set to define the visual interpretation grammar of the episode. Once the transformations are determined, we randomly assign a specific shape for the shape-based transformation, a specific color for the color-based transformation, and an indicator object for the neighbor-based transformation. Importantly, the indicator object is constrained to neither share the shape associated with the shape-based transformation nor the color linked to the color-based transformation.

Using these specifications, we generate input-output grid pairs representing primitive, level-1, and level-2 transformations. For each transformation mapping, we randomly place an object on a $10 \times 10$ grid, ensuring it possesses the designated shape, color, and/or proximity to the indicator object as required. The specified transformation is then applied to this object. If the resulting transformed object remains within the grid bounds and does not overlap with any other object, the corresponding input-output grid pair is accepted as a valid sample for the episode. Otherwise, a new object location is sampled and the process is repeated until a valid pair is obtained. Finally, we make sure that each

episode follows a unique grammar, i.e., that no two combinations of shape, color, and indicator objects correspond to the same set of transformations within the dataset.

Once the dataset is generated, we apply a systematicity-aware data split into training, validation, and test sets. As mentioned before, the five basic geometric transformations, along with three types of transformation indicators, allow us to define a total of ten distinct transformation triplets (e.g., shape-based: translation, color-based: reflection, neighbor-based: extension). We split the data as follows: we randomly designate 20% of all triplets as test-only and 80% as train-only (see Table 7). This means that the geometric transformations involved in the final query level-2 compositions differ between the training and evaluation sets. For instance, for seed 1860, all episodes whose triplet falls in the train set yield 82,908 training episodes, and evaluation-only triplets form a 17,092-episode pool, which we split evenly into 8,546 validation and 8,546 test episodes. All data is publicly available at: https://huggingface.co/datasets/mainlp/Compositional-ARC.

## B.4 DATASET STATISTICS

Table 7 presents detailed statistics for the datasets used in this study. As outlined in Section 5.1, we train and evaluate models via MLC across four distinct dataset splits to mitigate the influence of randomness in the data split process. The table includes the number of training, validation, and test samples for each split. Additionally, it provides information on the query transformation compositions present in the training and test sets, along with the frequency of each basic geometric transformation within the train dataset.

In a similar vein, Table 8 shows the statistics for the dataset version that includes more diverse transformations, as described in Section 5.3. The table provides information on the query transformation compositions present in the training and test sets, along with the frequency of each geometric transformation within the training dataset.

## C TRAINING DETAILS

As outlined in Section 3.2, we use a transformer-based encoder-decoder model trained using MLC to predict the correct output grid for a given input query, given a set of study examples. Specifically, we generate a dataset of 100,000 episodes and split it into train, validation and test sets (for more information see Section 4.1 and Table 7). The model is optimized using cross-entropy loss, averaged over the predicted patch embeddings, as described in Section 3.2. To place greater emphasis on non-background regions, patches corresponding exclusively to black $2 \times 2$ cells are down-weighted by a factor of $0.2$ during loss computation.

Each episode includes a collection of study examples and queries. In the standard few-shot learning task (Section 4.1), the model receives three input-output grid pairs, along with the input query. For the systematicity task, 12 systematic study examples are provided. In both tasks, the model is required to predict the correct output grid for ten distinct input queries.

Training is conducted over 200 epochs with a batch size of 200 for the standard few-shot learning task (i.e., $200 \cdot 10 = 2000$ queries per batch), and over 300 epochs with the same batch size for the systematicity task. A learning rate of 0.01 is used in both cases. Following the approach of Lake & Baroni (2023), we apply a warm-up phase during the first episode, beginning with a learning rate of $1 \times 10^{-4}$, followed by a linear decay to $5 \times 10^{-4}$ over the course of training. Additional hyperparameter settings are provided in Section C.1 and summarized in Table 3.

## C.1 HYPERPARAMETERS

To identify suitable hyperparameters for model training, we conduct Bayesian search over a predefined range of values: learning rate $\in [1 \times 10^{-2}, 1 \times 10^{-3}, 1 \times 10^{-4}]$, final learning rate after linear decay $\in [1 \times 10^{-4}, 5 \times 10^{-4}]$, dropout rate $\in [0.0, 0.1, 0.2]$, gradient accumulation over $k \in [1, 2]$ batches, cell color perturbation probability $p_{\text{noise}} \in [0.0, 0.01, 0.001]$, feedforward hidden dimension $\in [512, 768]$, loss weighting for background (all-black) patches $\in [0.2, 0.4, 1.0]$, number of encoder layers $\in [2, 3, 4]$, and number of decoder layers $\in 2, 3, 4$.

Table 3: Hyperparameter configuration for models trained via MLC.

| Parameter | Value | Parameter | Value |
|---|---|---|---|
| number layers in decoder | 3 | learning rate after training | $5 \times 10^{-4}$ |
| number layers in decoder | 3 | dropout | 0.0 |
| number of attention heads | 8 | weight decay | 0.01 |
| hidden dimension | 128 | noise probability | 0.001 |
| feedforward hidden size | 768 | gradient accumulation over $k$ batches | 2 |
| learning rate | 0.01 | background patch loss weight | 0.2 |

For the hyperparamter search, the model is trained for 40 epochs on the systematicity task and evaluated on its corresponding validation set. Across 25 independent runs, we select the configuration that achieves the highest validation accuracy. The final hyperparameter settings, presented in Table 3, are employed consistently across both task setups.

## C.2 Implementation details

All experiments were conducted using PyTorch (Paszke et al., 2019) as the primary development framework. Comprehensive details regarding supporting software and versioning are available in our code repository. Experiments were executed on NVIDIA A100 and H200 GPUs. Training models with MLC on the standard three-shot learning task over 200 epochs required approximately 40 GPU hours on a single A100 GPU. For the systematicity experiments with 12 study examples, training over 300 epochs on the designated dataset consumed roughly 100 GPU hours on a single H200 GPU.

## C.3 Original MLC training

In the original MLC setup, Lake & Baroni (2023) train a standard seq2seq transformer (3-layer encoder/decoder, 8 attention heads, hidden dimension 128) with Adam on 100,000 dynamically generated episodes of pseudo-language instructions. Each episode is defined by a latent compositional grammar and contains multiple study examples and queries concatenated into a single input sequence, as described in Section 2. Training minimizes token-level cross-entropy for 50 epochs using a batch size of 25 episodes, a learning rate of $10^{-3}$ with a one-epoch warm-up followed by linear decay, and dropout of 0.1. Compared to our approach, their model targets linguistic sequences rather than visual grids and does not use patch-wise losses or background reweighting. Additionally, we train longer with larger batches and a loss designed to emphasize non-background spatial structure.

# D Experiment details

This section provides further details regarding our experimental setup. Specifically, Section D.1 presents formal definitions of the evaluation metrics, while Section D.2 outlines additional information on how we interact with API-based LLMs.

## D.1 Evaluation metrics

As described in Section 4.3, we use three different evaluation metrics to assess model performance: i) exact match accuracy, ii) color accuracy, and iii) shape accuracy. These metrics are formally defined based on the grid-based environment $\boldsymbol{X}$ and the concept of an object $\mathbb{O}$, as specified in Definition 2.

Let $\boldsymbol{X}^{target}, \boldsymbol{X}^{pred} \in \mathbb{N}^{10 \times 10}$ denote the target and predicted grids, respectively. Each cell $\boldsymbol{X}_{ij}^{target}$ (or $\boldsymbol{X}_{ij}^{pred}$) contains an integer in $0, \dots, 9$, where 0 represents the background and values from 1 to 9 correspond to cells occupied by colored objects. The set of objects—defined as maximal connected cells of a consistent color under the Moore neighborhood (see Section 3.1)—extracted from $\boldsymbol{X}^{target}$ and $\boldsymbol{X}^{pred}$ are denoted $\mathcal{P}(\boldsymbol{X}^{target})$ and $\mathcal{P}(\boldsymbol{X}^{pred})$, respectively. For each object in grid $\mathbb{O} \in \mathcal{P}(\boldsymbol{X})$, we assign a color label $c(\mathbb{O}) \in 1, \dots, 9$ and define its normalized shape as follows:

$$S(\mathbb{O}) = \{(i - i_{\min}, \, j - j_{\min}) : (i,j) \in \mathbb{O}\}, \tag{1}$$

where

$$i_{\min} = \min\{i : (i,j) \in \mathbb{O}\} \quad \text{and} \quad j_{\min} = \min\{j : (i,j) \in \mathbb{O}\}. \tag{2}$$

This transformation "anchors" the object to the top-left corner by translating it to a coordinate system with its minimum row and column indices set to zero.

**Accuracy.** The exact match accuracy evaluates whether the predicted grid $\boldsymbol{X}^{pred}$ is identical to the target grid $\boldsymbol{X}^{target}$ on a cell-by-cell basis:

$$\text{Accuracy}(\boldsymbol{X}^{pred}, \boldsymbol{X}^{target}) = \begin{cases} 1, & \text{if } \boldsymbol{X}_{ij}^{pred} = \boldsymbol{X}_{ij}^{target} \quad \forall \, (i,j) \in \{0, \dots, 9\}^2, \\ 0, & \text{otherwise.} \end{cases} \tag{3}$$

In other words, this metric yields a value of 1 if and only if the entire predicted grid matches the target grid exactly, i.e., $\boldsymbol{X}^{target} = \boldsymbol{X}^{pred}$. The mean accuracy over the dataset $\mathcal{D}$ is then defined as:

$$\text{Accuracy} = \frac{1}{|\mathcal{D}|} \sum_{(\boldsymbol{X}^{pred}, \boldsymbol{X}^{target}) \in \mathcal{D}} \text{Accuracy}(\boldsymbol{X}^{pred}, \boldsymbol{X}^{target}) \tag{4}$$

**Color accuracy.** Color accuracy assesses whether the predicted grid contains the same number of objects of each color as the target grid, irrespective of their locations or shapes. For a given color $c \in 1, \dots, 9$, let

$$m(c, \boldsymbol{X}) = \big|\{\mathbb{O} \in \mathcal{P}(\boldsymbol{X}) : c(\mathbb{O}) = c\}\big|. \tag{5}$$

denote the number of objects of color $c$ in grid $X$. Then, *color accuracy* is defined as:

$$\text{Color Accuracy}(\boldsymbol{X}^{pred}, \boldsymbol{X}^{target}) = \mathbb{1}\Big\{\forall \, c \in \{1, \dots, 9\} : \, m(c, \boldsymbol{X}^{pred}) = m(c, \boldsymbol{X}^{target})\Big\}, \tag{6}$$

where $\mathbb{1}\cdot$ is the indicator function, returning 1 if the condition is satisfied for all colors and 0 otherwise. The mean color accuracy over the dataset $\mathcal{D}$ is given by:

$$\text{Color Accuracy} = \frac{1}{|\mathcal{D}|} \sum_{(\boldsymbol{X}^{pred}, \boldsymbol{X}^{target}) \in \mathcal{D}} \text{Color Accuracy}(\boldsymbol{X}^{pred}, \boldsymbol{X}^{target}) \tag{7}$$

**Shape accuracy.** Shape accuracy measures the agreement in object shapes between the predicted and target grids, independent of color and position. For each object in a grid $\mathbb{O} \in \mathcal{P}(\boldsymbol{X})$, we consider its normalized shape $S(\mathbb{O})$ as defined in Equation 1. The count of objects with a specific normalized shape $s$ in grid $\boldsymbol{X}$ is given by:

$$n(s, \boldsymbol{X}) = \big|\{\mathbb{O} \in \mathcal{P}(\boldsymbol{X}) : S(\mathbb{O}) = s\}\big|. \tag{8}$$

Accordingly, *shape accuracy* is defined as:

$$\text{Shape Accuracy}(\boldsymbol{X}^{pred}, \boldsymbol{X}^{target}) = \mathbb{1}\Big\{\forall \, s : \, n(s, \boldsymbol{X}^{pred}) = n(s, \boldsymbol{X}^{target})\Big\}. \tag{9}$$

That is, the predicted grid $\boldsymbol{X}^{pred}$ has perfect shape accuracy if the number of objects corresponding to each normalized shape is identical to that in the target grid $\boldsymbol{X}^{target}$. Finally, the mean shape accuracy over the dataset $\mathcal{D}$ is given by:

$$\text{Shape Accuracy} = \frac{1}{|\mathcal{D}|} \sum_{(\boldsymbol{X}^{pred}, \boldsymbol{X}^{target}) \in \mathcal{D}} \text{Shape Accuracy}(\boldsymbol{X}^{pred}, \boldsymbol{X}^{target}) \tag{10}$$

## D.2 MODEL INFORMATION

**General-purpose LLMs.** As described in Section 4.2, we evaluate three different general-purpose LLMs on *Compositional-ARC*. Specifically, we assess the performance of o3-mini (OpenAI, 2025) (version o3-mini-2025-01-31[3]), GPT-4o (Achiam et al., 2023) (version gpt-4o-2024-08-06[4]), and Gemini 2.0 Flash (DeepMind, 2024) (version gemini-2.0-flash-001[5]). All models are accessed via their respective batch APIs, enabling us to process multiple samples per request. Unless otherwise specified, we employ the default API settings. For GPT-4o and o3-mini, this corresponds to a temperature and top_p value of $1.0$.[6] Due to financial constraints, the o3-mini model is configured with a "low" reasoning effort. For Gemini 2.0 Flash, the provider does not disclose default parameter settings.

**Prompts.** The complete set of prompts used in our evaluations is presented in Figures 12 through 15. To ensure consistency and facilitate meaningful comparisons, we apply the same prompts across all models. The standard few-shot learning prompt appears in Figure 12, while the prompt used for the systematicity task is shown in Figure 14. For Gemini 2.0 Flash, we add the instruction: "Do not generate any code to solve the task" to the output requirements, as the model otherwise does not adhere to the required output format. As outlined in Section 4.2, we additionally evaluate GPT-4o and Gemini 2.0 Flash in a multimodal configuration, in which both an image of the study examples and the input query are provided alongside the text prompt (text+image). The multimodal prompt for the few-shot learning task is shown in Figure 13, with the accompanying image illustrated in Figure 10. The corresponding multimodal prompt for the systematicity task is depicted in Figure 15, with the associated image presented in Figure 11. For the textual prompts, we represent grids as two-dimensional arrays, consistent with prior work (Moskvichev et al., 2023)). For instance, the final query input grid in Figure 5 would be represented as:

```
[[0, 0, 0, 0, 0, 0, 0, 0, 0, 0],
 [0, 0, 0, 0, 0, 0, 0, 0, 0, 0],
 [0, 0, 0, 0, 0, 0, 0, 0, 0, 0],
 [0, 0, 0, 0, 0, 0, 0, 0, 0, 0],
 [0, 5, 0, 0, 0, 0, 0, 0, 0, 0],
 [0, 5, 0, 0, 0, 0, 0, 0, 0, 0],
 [5, 5, 0, 0, 0, 0, 0, 0, 0, 0],
 [0, 5, 0, 0, 0, 0, 0, 0, 0, 0],
 [0, 0, 0, 0, 1, 0, 0, 0, 0, 0],
 [0, 0, 0, 0, 1, 1, 0, 0, 0, 0]]
```

Model responses are parsed using regular expressions to identify the expression "output:", followed by a two-dimensional array of the form "[[...]]", as specified in the input prompt. If a response does not contain this pattern, it is excluded from further analysis and omitted from accuracy computations. Table 4 summarizes the proportion of valid responses for each model.

**Domain-specific LLMs.** As mentioned in Section 4.2, we also evaluate two LLMs proposed by Franzen et al. (2024) that are specifically tailored to ARC-style data: (i) Llama-3.2-3B-ReARC (version Llama-3.2-3B-ARChitects-ReArc-bnb-4bit[7]) and (ii) Mistral-NeMO-Minitron-8B-Full (version Mistral-NeMo-Minitron-8B-ARChitects-Full-bnb-4bit[8]). We use the original code[9] provided by the authors to run their models on *Compositional-ARC*, with default parameters. This means that the models perform augmented inference on the test set with rotations and transpositions over all symmetries, in addition to color permutations and example

---

[3]https://platform.openai.com/docs/models/o3-mini

[4]https://platform.openai.com/docs/models/gpt-4o

[5]https://ai.google.dev/gemini-api/docs/models#gemini-2.0-flash

[6]https://platform.openai.com/docs/api-reference/chat/create

[7]https://huggingface.co/da-fr/Llama-3.2-3B-ARChitects-ReArc-bnb-4bit

[8]https://huggingface.co/da-fr/Mistral-NeMo-Minitron-8B-ARChitects-Full-bnb-4bit

[9]https://github.com/da-fr/arc-prize-2024

shuffling. Candidate pruning is further applied with a minimum probability of 0.1. For models evaluated with test-time training, we follow the authors' one-epoch LoRA adaptation on the study examples of the test data repeated 48 times with the same augmentations described before. LoRA targets the attention and MLP modules, as well as the embeddings, with $r = 64$, $\alpha = 16$, and dropout set to 0. The models are trained with a batch size of 16, gradient accumulation set to 1, a cosine learning rate of $1 \times 10^{-4}$ (with $1 \times 10^{-5}$ for embeddings), and a warmup ratio of 0.25. The resulting weights are then used for inference with the same default settings as described earlier.

# E  ADDITIONAL RESULTS

In this section, we present additional results for the experiments conducted in this study. First, we present additional qualitative results related to the model predictions on the standard few-shot learning and the systematicity task. Figures 5 through 7 illustrate representative episodes from the standard few-shot learning task. Model predictions are shown adjacent to each query, with results for GPT-4o and Gemini 2.0 Flash corresponding to text-only prompts. Across all three episodes, the model trained using MLC consistently predicts the correct output grid. In contrast, GPT-4o and Gemini 2.0 Flash frequently fail to identify the correct transformation—either misrepresenting the shape of the transformed object or incorrectly predicting its final position. Notably, o3-mini successfully predicts the correct output for the episodes in Figures 6 and 7, but fails on the example in Figure 5. Figures 8 and 9 highlight episodes from the systematicity task. As shown, all general-purpose LLMs fail to produce accurate transformations, often misplacing the transformed object within the grid. In contrast, the model trained via MLC consistently predicts the correct transformation.

**Response rates.**    As outlined in Section D.2, the general-purpose LLMs we evaluate are instructed to present their final output grid predictions using the keyword "output:", followed by a two-dimensional array of size $10 \times 10$ in the format "[[...]]". Responses that do not conform to this expected pattern are excluded from subsequent analyses and are not included in accuracy calculations. Table 4 provides an overview of the proportion of valid responses for each model. In the standard few-shot learning setting, all models demonstrate very high valid response rates, exceeding 99%. However, in the systematicity task, a slight decrease in valid responses is observed for Gemini 2.0 Flash when additional visual input (text+image) is introduced, with the rate falling to 94.09%. More significantly, GPT-4o exhibits a notable drop in valid response rate to 77.24% under multimodal conditions. We hypothesize that this decline may be attributed to the increased context length resulting from the additional image input.

**Error Analysis.**    As described in Section 5.2, we analyze the models' predictions and compare them with common failure modes. Table 5 shows the percentage of each error type described in Section 5.2 across models. For errors related to the models predicting a primitive or level-1 transformation instead of the desired level-2 transformation composition, we further illustrate which specific primitive or level-1 transformation was applied in Table 6. Specifically, this table shows whether the primitive

Table 4: The proportion of valid responses generated by the different models reported for the standard three-shot learning task and the systematicity task. For general-purpose LLMs, valid responses must contain the string "output:", followed by a two-dimensional $10 \times 10$ array of the form "[[...]]".

| Model | Valid Responses (3-Shot) | Valid Responses (Systematicity) |
|---|---|---|
| GPT-4o | 99.95% | 99.40% |
| + *image* | 99.80% | 77.24% |
| Gemini 2.0 Flash | 99.92% | 99.74% |
| + *image* | 99.51% | 94.09% |
| o3-mini (low) | 100% | 100% |
| Llama-3.2-3B-ReARC | 100% | 100% |
| + *test-time training* | - | 100% |
| Mistral-NeMO-Minitron-8B-Full | 100% | 100% |
| + *test-time training* | - | 100% |
| MLC (ours) | 100% | 100% |

Table 5: Error distribution by error category across models. Values denote the percentage (%) of prediction errors assigned to each error category.

| Model | Format | No Transform | Primitive | Level-1 | Invalid Position | Invalid Shape | Other |
|---|---|---|---|---|---|---|---|
| GPT-4o | 0.60 | 0.46 | 4.59 | 7.71 | 6.62 | 79.26 | 0.77 |
| + *image* | 22.91 | 3.10 | 4.19 | 4.09 | 4.26 | 59.84 | 1.60 |
| Gemini 2.0 Flash | 0.26 | 1.56 | 11.41 | 22.41 | 5.73 | 58.32 | 0.30 |
| + *image* | 7.60 | 0.72 | 9.05 | 15.52 | 4.60 | 61.83 | 0.68 |
| o3-mini (low) | 0.00 | 5.06 | 30.86 | 13.08 | 0.79 | 49.31 | 0.91 |
| Llama-3.2-3B-ReARC | 0.00 | 55.28 | 25.13 | 4.77 | 0.30 | 14.47 | 0.06 |
| + *test-time training* | 0.00 | 0.44 | 6.18 | 67.13 | 10.72 | 15.52 | 0.00 |
| Mistral-NeMO-Minitron-8B-Full | 0.00 | 0.54 | 53.41 | 28.89 | 0.12 | 17.03 | 0.01 |
| + *test-time training* | 0.00 | 2.47 | 5.05 | 71.55 | 13.31 | 7.62 | 0.00 |
| MLC (ours) | 0.05 | 3.34 | 3.56 | 9.11 | 3.02 | 70.57 | 10.35 |

transformation applied was based on the object's shape, color, or neighboring object. Similarly, the table illustrates which specific level-1 transformation composition was applied.

**Training on static data.** In addition to the model trained via MLC on a stream of dynamically changing visual interpretation grammars, as described in Section 3.2, we adopt the approach of Lake (2019) and train a transformer-based encoder-decoder on a dataset governed by a fixed visual grammar (referred to as *basic seq2seq*). This means that the indicator-transformation mappings are static across the whole dataset. For instance, if yellow objects translate one step downward, then this applies to all data samples across the dataset. Instead of episodes with few-shot examples, this dataset comprises individual input-output grid pairs, where the objective is to predict the output grid corresponding to a given input grid. This more closely resembles a standard training approach.

We construct a dataset of 1,300 grid pairs, partitioned into 1,260 training samples, 20 validation samples, and 20 test samples. Samples represent primitive transformations, as well as level-1 and level-2 transformation compositions. As with our other experiments, the test set includes level-2 transformation compositions that were not observed during training—only their constituent components and level-1 compositions were seen during training. For instance, the test set might include transformations composed of shape-based downward translation, color-based horizontal reflection, and neighbor-based upward extension. However, only their decomposed elements have been shown during training.

The model is trained for 200 epochs on the dataset using the parameters specified in Section C. While it successfully fits the training data (with an accuracy of over 99%), it fails to generalize to the out-of-distribution test set, achieving a test accuracy of 0.0%. This demonstrates that traditional model training, sample by sample, does not encourage systematic generalization to unseen composi-

Table 6: Percentages of errors falling into each primitive and level-1 transformation error category.

| Model | Primitive Transformations | | | Level-1 Transformations | | |
|---|---|---|---|---|---|---|
| | Shape | Color | Neighbor | Shape+Color | Shape+Neighbor | Color+Neighbor |
| GPT-4o | 2.09 | 1.56 | 0.93 | 4.68 | 2.10 | 0.92 |
| + *image* | 1.97 | 1.57 | 0.66 | 2.65 | 1.01 | 0.42 |
| Gemini 2.0 Flash | 4.22 | 4.70 | 2.49 | 16.17 | 4.05 | 2.19 |
| + *image* | 2.96 | 3.66 | 2.43 | 9.58 | 3.45 | 2.49 |
| o3-mini (low) | 16.67 | 13.07 | 1.12 | 11.16 | 0.99 | 0.93 |
| Llama-3.2-3B-ReARC | 14.22 | 6.48 | 4.43 | 2.80 | 1.74 | 0.24 |
| + *test-time training* | 0.27 | 5.16 | 0.76 | 8.81 | 35.68 | 22.64 |
| Mistral-NeMO-Minitron-8B-Full | 39.44 | 7.46 | 6.50 | 15.30 | 13.49 | 0.11 |
| + *test-time training* | 0.05 | 4.08 | 0.91 | 8.05 | 38.81 | 24.69 |
| MLC (ours) | 0.16 | 1.08 | 2.32 | 5.98 | 2.16 | 0.97 |

tions. Instead, systematicity requires a training procedure with examples over dynamically varying interpretation grammars, as described in Section 3.2.

## F  USE OF AI ASSISTANTS

We used GitHub Copilot for parts of the project's code, and ChatGPT for minor language revisions.

Table 7: Summary of dataset statistics across different dataset splits, each determined by a distinct random seed. Listed are the number of episodes in the training, validation, and test sets. Additionally, the final query transformation compositions (level 2) are reported for both the training and evaluation datasets. The rightmost column details the frequency of each basic geometric transformation present in the training dataset.

| Data Split | No. Episodes | | Query Transformations | | Basic Transformations | |
|---|---|---|---|---|---|---|
| | Set | No. | Type | Composition | Transformation | Freq. |
| seed 1860 | Train | 82908 | | translation+reflection+coloring | red coloring | 35828 |
| | Val | 8546 | | reflection+rotation+extension | orange coloring | 35819 |
| | Test | 8546 | | translation+reflection+rotation | down translation | 23398 |
| | | | Train | translation+rotation+coloring | right translation | 27021 |
| | | | | reflection+coloring+extension | leftward extension | 22140 |
| | | | | reflection+rotation+coloring | upward extension | 21806 |
| | | | | translation+coloring+extension | cw. rotation | 19551 |
| | | | | rotation+coloring+extension | ccw. rotation | 19394 |
| | | | Test | translation+rotation+extension | horizontal reflection | 21967 |
| | | | | translation+reflection+extension | vertical reflection | 21800 |
| seed 1870 | Train | 83481 | | translation+rotation+extension | red coloring | 27603 |
| | Val | 8259 | | translation+reflection+rotation | orange coloring | 27525 |
| | Test | 8260 | | reflection+rotation+extension | down translation | 31385 |
| | | | Train | reflection+coloring+extension | right translation | 36126 |
| | | | | translation+reflection+extension | leftward extension | 26501 |
| | | | | translation+rotation+coloring | upward extension | 25913 |
| | | | | translation+reflection+coloring | cw. rotation | 15421 |
| | | | | translation+coloring+extension | ccw. rotation | 15283 |
| | | | Test | rotation+coloring+extension | horizontal reflection | 22366 |
| | | | | reflection+rotation+coloring | vertical reflection | 22320 |
| seed 1880 | Train | 80035 | | translation+coloring+extension | red coloring | 25850 |
| | Val | 9982 | | translation+rotation+extension | orange coloring | 25832 |
| | Test | 9983 | | translation+rotation+coloring | down translation | 31385 |
| | | | Train | reflection+rotation+extension | right translation | 36126 |
| | | | | translation+reflection+coloring | leftward extension | 24821 |
| | | | | translation+reflection+extension | upward extension | 24147 |
| | | | | translation+reflection+rotation | cw. rotation | 19734 |
| | | | | rotation+coloring+extension | ccw. rotation | 19594 |
| | | | Test | reflection+rotation+coloring | horizontal reflection | 16331 |
| | | | | reflection+coloring+extension | vertical reflection | 16285 |
| seed 1890 | Train | 80557 | | translation+coloring+extension | red coloring | 30227 |
| | Val | 9721 | | translation+reflection+rotation | orange coloring | 30255 |
| | Test | 9722 | | rotation+coloring+extension | down translation | 23279 |
| | | | Train | translation+reflection+coloring | right translation | 24789 |
| | | | | reflection+rotation+extension | leftward extension | 26483 |
| | | | | translation+reflection+extension | upward extension | 26277 |
| | | | | reflection+coloring+extension | cw. rotation | 13949 |
| | | | | reflection+rotation+coloring | ccw. rotation | 13831 |
| | | | Test | translation+rotation+coloring | horizontal reflection | 26329 |
| | | | | translation+rotation+extension | vertical reflection | 26252 |

Table 8: Statistics of the dataset version including more diverse transformations. Listed are the number of episodes in the training, validation, and test sets. Additionally, the final query transformation compositions (level 2) are reported for both the training and evaluation datasets. The rightmost column details the frequency of each basic geometric transformation present in the training dataset.

| Data Split | No. Episodes | | Query Transformations | | Basic Transformations | |
|---|---|---|---|---|---|---|
| | Set | No. | Type | Composition | Transformation | Freq. |
| seed 1860 | Train | 85528 | Train | translation+reflection+coloring | red coloring | 18376 |
| | Val | 5472 | | reflection+rotation+extension | orange coloring | 18627 |
| | Test | 5473 | | translation+reflection+rotation | yellow coloring | 18961 |
| | | | | translation+rotation+coloring | green coloring | 18491 |
| | | | | reflection+coloring+extension | 1-step left translation | 6471 |
| | | | | reflection+rotation+coloring | 2-step left translation | 3671 |
| | | | | translation+coloring+extension | 1-step right translation | 7942 |
| | | | | rotation+coloring+extension | 2-step right translation | 5438 |
| | | | Test | translation+rotation+extension | 1-step up translation | 6780 |
| | | | | translation+reflection+extension | 2-step up translation | 4051 |
| | | | | | 1-step down translation | 6686 |
| | | | | | 2-step down translation | 4022 |
| | | | | | leftward extension | 11742 |
| | | | | | rightward extension | 12071 |
| | | | | | upward extension | 11801 |
| | | | | | downward extension | 12909 |
| | | | | | cw. rotation | 20797 |
| | | | | | ccw. rotation | 20799 |
| | | | | | horizontal reflection | 23536 |
| | | | | | vertical reflection | 23413 |

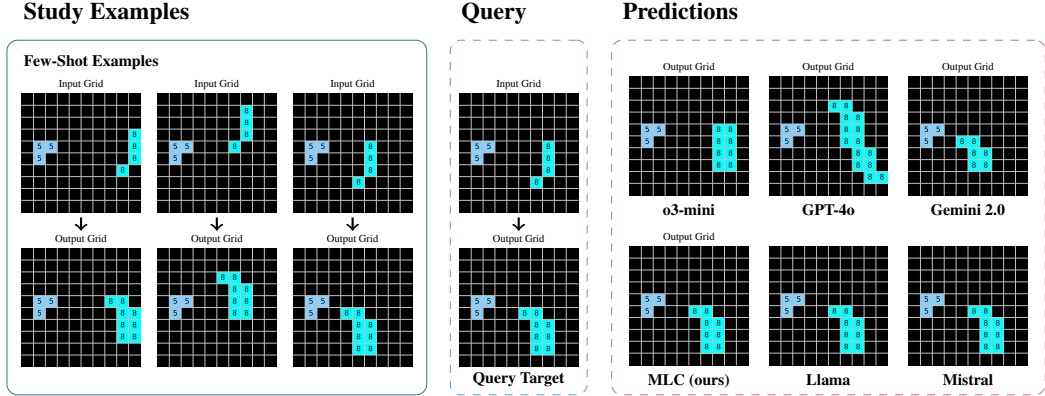

Figure 5: An example of the few-shot learning task. Models are provided with three study examples that demonstrate the transformation that needs to be inferred for the final input grid. Model predictions are displayed to the right.

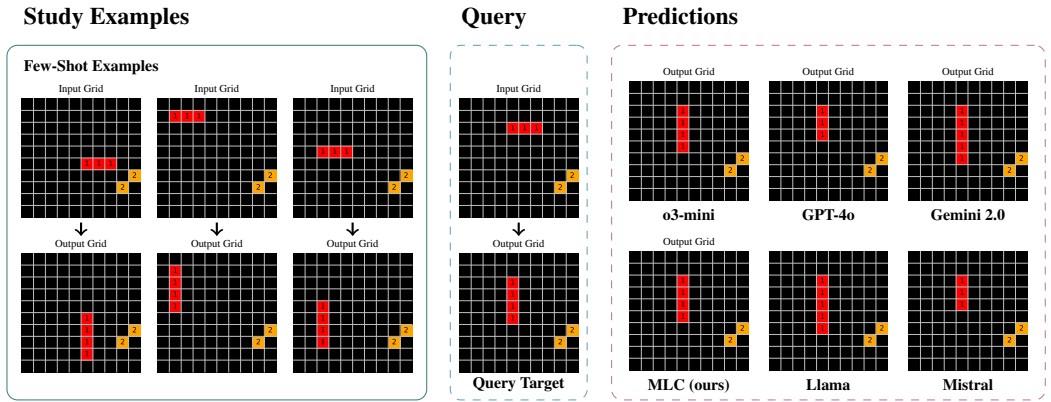

Figure 6: A second example of the few-shot learning task. Models are provided with three study examples that demonstrate the transformation that needs to be inferred for the final input grid. Model predictions are displayed to the right.

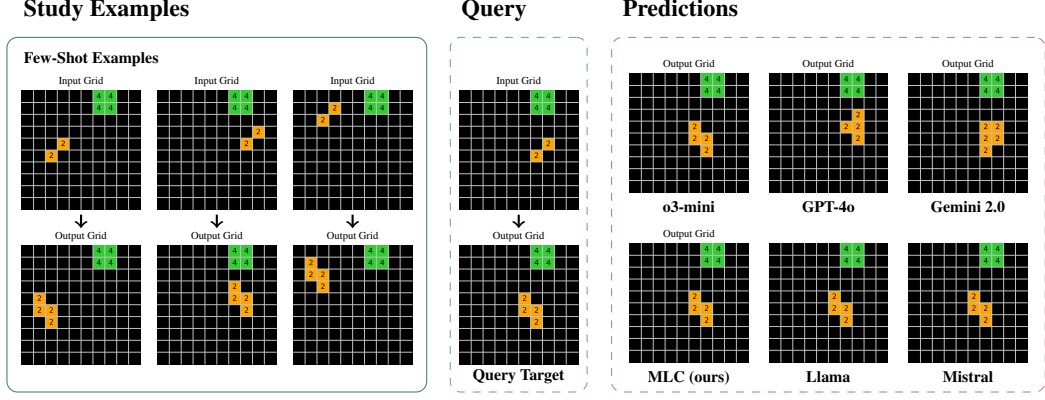

Figure 7: A third example of the few-shot learning task. Models are provided with three study examples that demonstrate the transformation that needs to be inferred for the final input grid. Model predictions are displayed to the right.

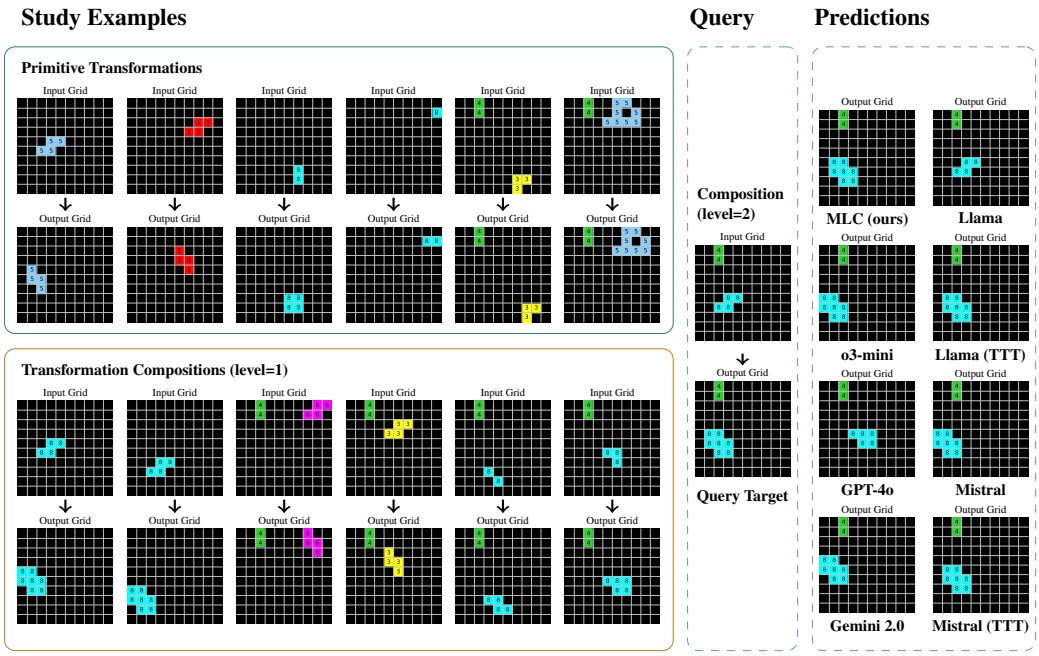

Figure 8: An episode from the systematicity task. Given a set of study examples comprising primitive transformations and level-1 transformation compositions, models are asked to predict the output grid for a previously unseen level-2 transformation composition. Predictions of different models are presented to the right.

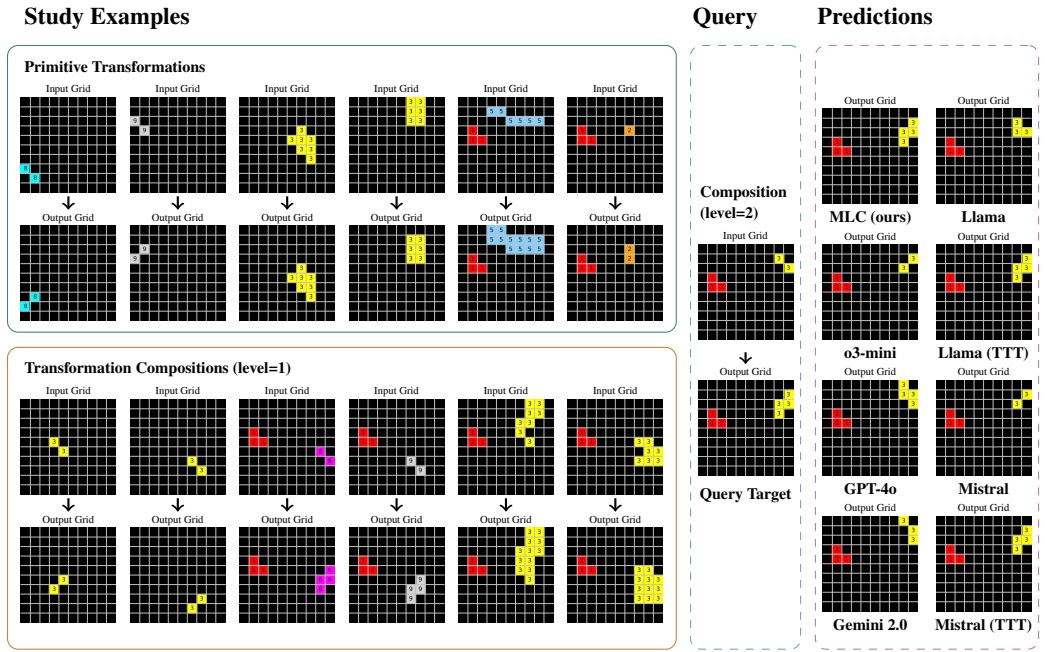

Figure 9: Another episodes from the systematicity task. Given a set of study examples comprising primitive transformations and level-1 transformation compositions, models are asked to predict the output grid for a previously unseen level-2 transformation composition. Predictions of different models are presented to the right.

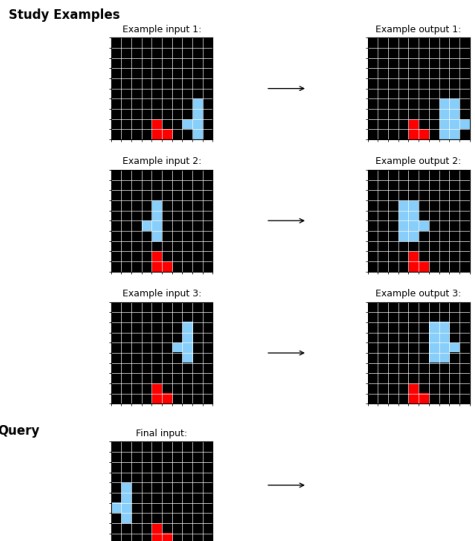

Figure 10: An exemplary visual input used in the multimodal prompt for the 3-shot learning task.

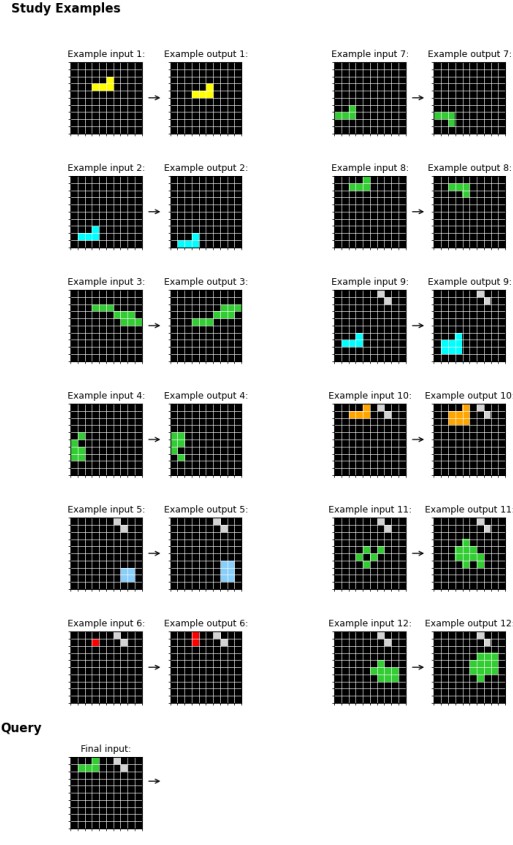

Figure 11: An exemplary visual input used in the multimodal prompt for the systematicity task.

---

**Text-Only 3-Shot Prompt**

### Task Description:
You must solve an abstract visual reasoning task by identifying geometric transformations (e.g., rotation, translation, color changes, etc.) applied to objects within a 10x10 grid.

To infer the correct geometric transformation, you are given a series of **3 pairs of input-output examples**. Each example pair consists of:

- An **input grid**: a 10x10 list of lists (2d array), where each element is an integer (0-9).
- A corresponding **output grid**: a 10x10 list of lists (2d array) that has undergone a transformation based on a specific geometric rule.

For the prediction you need to understand the transformations displayed in the provided examples and apply them to the final input grid.

### Your Task:
1. **Analyze** the example pairs to infer the transformation rules applied to each input grid.
2. **Identify** how these transformations are applied to generate the output grids.
3. **Apply** the deduced transformations to the final input grid.
4. **Output** the correctly transformed 10x10 grid.

### Output Requirements:
- **Return only the final output grid.**
- Do not include any extra text, explanations, or comments.
- The output must be formatted exactly as: 'output: [[...]]'
- The output grid must be a 10x10 list of lists containing only integers between 0 and 9 (inclusive).
- Do not include unnecessary line breaks or additional text beyond the specified format.

### Input Format:
You will receive the following data:

1. **Study examples:** A list of 3 few-shot example pairs, formatted as:
   'example input 1: [[...]], example output 1: [[...]], ..., example input 3: [[...]], example output 3: [[...]]'
2. **Final input:** A single 10x10 list of lists on which you must apply the inferred transformation(s).

Your goal is to determine the correct transformation and return the final output grid.

### Input:
Study examples:
example input 1: <2-dimensional array representing the input grid of example 1>
example output 1: <2-dimensional array representing the output grid of example 1>
...
example input 3: <2-dimensional array representing the input grid of example 3>
example output 3: <2-dimensional array representing the output grid of example 3>

Final input: <2-dimensional array representing the final query input grid>

Figure 12: The prompt used for the few-shot experiment when instructing LLMs in (text-only) mode. Text enclosed in sharp brackets $<\ldots>$ is replaced by the actual examples.

Text+Image 3-Shot Prompt

### Task Description:
You must solve an abstract visual reasoning task by identifying geometric transformations (e.g., rotation, translation, color changes, etc.) applied to objects within a 10x10 grid.

To infer the correct geometric transformation, you are given a series of **3 pairs of input-output examples**. Each example pair consists of:

- An **input grid**: a 10x10 list of lists (2d array), where each element is an integer (0-9).
- A corresponding **output grid**: a 10x10 list of lists (2d array) that has undergone a transformation based on a specific geometric rule.

For the prediction you need to understand the transformations displayed in the provided examples and apply them to the final input grid.

### Your Task:
1. **Analyze** the example pairs to infer the transformation rules applied to each input grid.
2. **Identify** how these transformations are applied to generate the output grids.
3. **Apply** the deduced transformations to the final input grid.
4. **Output** the correctly transformed 10x10 grid.

### Output Requirements:
- **Return only the final output grid.**
- Do not include any extra text, explanations, or comments.
- The output must be formatted exactly as: 'output: [[...]]'
- The output grid must be a 10x10 list of lists containing only integers between 0 and 9 (inclusive).
- Do not include unnecessary line breaks or additional text beyond the specified format.

### Input Format:
You will receive the following data:
1. **Study examples:** A list of 3 few-shot example pairs, formatted as:
   'example input 1: [[...]], example output 1: [[...]], ..., example input 3: [[...]], example output 3: [[...]]'
2. **Final input:** A single 10x10 list of lists on which you must apply the inferred transformation(s).
3. **Image input:** Additionally, you receive an image that visualizes the 3 few-shot example pairs and the final input query.

Your goal is to determine the correct transformation and return the final output grid.

### Input:
Study examples:
example input 1: <2-dimensional array representing the input grid of example 1>
example output 1: <2-dimensional array representing the output grid of example 1>
...
example input 3: <2-dimensional array representing the input grid of example 3>
example output 3: <2-dimensional array representing the output grid of example 3>

Final input: <2-dimensional array representing the final query input grid>

Figure 13: The prompt used for the few-shot experiment when instructing LLMs in (text+image) mode. Text enclosed in sharp brackets $< \ldots >$ is replaced by the actual examples. Additionally, the model is provided with the image in Figure 10.

---

**Text-Only Systematicity Prompt**

### Task Description:
You must solve an abstract visual reasoning task by identifying geometric transformations (e.g., rotation, translation, color changes, etc.) applied to objects within a 10x10 grid.

To infer the correct geometric transformation, you are given a series of **12 pairs of input-output examples**. Each example pair consists of:
- An **input grid**: a 10x10 list of lists (2d array), where each element is an integer (0-9).
- A corresponding **output grid**: a 10x10 list of lists (2d array) that has undergone a transformation based on a specific geometric rule.

The first 6 example pairs demonstrate primitive transformations based on the object's color, shape, or the presence of an additional object. For instance, objects of a certain color within the 10x10 input grid might undergo a translation, while objects of a certain shape (distinct numerical pattern) are being rotated.

The latter 6 example pairs involve **composite transformations**, meaning multiple transformations are applied simultaneously. For instance, for objects that have the appropriate color **and** shape, both a translation and rotation are applied simultaneously.

For the final prediction you need to understand and further combine the transformations displayed in the provided examples and apply them to the final input grid.

### Your Task:
1. **Analyze** the example pairs to infer the transformation rules applied to each input grid.
2. **Identify** how these transformations might combine to generate the output grids.
3. **Apply** the deduced transformations to the final input grid.
4. **Output** the correctly transformed 10x10 grid.

### Output Requirements:
- **Return only the final output grid.**
- Do not include any extra text, explanations, or comments.
- The output must be formatted exactly as: 'output: [[...]]'
- The output grid must be a 10x10 list of lists containing only integers between 0 and 9 (inclusive).
- Do not include unnecessary line breaks or additional text beyond the specified format.

### Input Format:
You will receive the following data:
1. **Study examples:** A list of 12 study example pairs, formatted as:
   'example input 1: [[...]], example output 1: [[...]], ..., example input 12: [[...]], example output 12: [[...]]'
2. **Final input:** A single 10x10 list of lists on which you must apply the inferred transformation(s).

Your goal is to determine the correct transformation and return the final output grid.

### Input:
Study examples:
example input 1: <2-dimensional array representing the input grid of example 1>
example output 1: <2-dimensional array representing the output grid of example 1>
...
example input 12: <2-dimensional array representing the input grid of example 12>
example output 12: <2-dimensional array representing the output grid of example 12>

Final input: <2-dimensional array representing the final query input grid>

Figure 14: The prompt used for the systematicity experiment when instructing LLMs in (text-only) mode. Text enclosed in sharp brackets < . . . > is replaced by the actual examples.

---

**Text+Image Systematicity Prompt**

### Task Description:
You must solve an abstract visual reasoning task by identifying geometric transformations (e.g., rotation, translation, color changes, etc.) applied to objects within a 10x10 grid.

To infer the correct geometric transformation, you are given a series of **12 pairs of input-output examples**. Each example pair consists of:

- An **input grid**: a 10x10 list of lists (2d array), where each element is an integer (0-9).
- A corresponding **output grid**: a 10x10 list of lists (2d array) that has undergone a transformation based on a specific geometric rule.

The first 6 example pairs demonstrate primitive transformations based on the object's color, shape, or the presence of an additional object. For instance, objects of a certain color within the 10x10 input grid might undergo a translation, while objects of a certain shape (distinct numerical pattern) are being rotated.

The latter 6 example pairs involve **composite transformations**, meaning multiple transformations are applied simultaneously. For instance, for objects that have the appropriate color **and** shape, both a translation and rotation are applied simultaneously.

For the final prediction you need to understand and further combine the transformations displayed in the provided examples and apply them to the final input grid.

### Your Task:
1. **Analyze** the example pairs to infer the transformation rules applied to each input grid.
2. **Identify** how these transformations might combine to generate the output grids.
3. **Apply** the deduced transformations to the final input grid.
4. **Output** the correctly transformed 10x10 grid.

### Output Requirements:
- **Return only the final output grid.**
- Do not include any extra text, explanations, or comments.
- The output must be formatted exactly as: 'output: [[...]]'
- The output grid must be a 10x10 list of lists containing only integers between 0 and 9 (inclusive).
- Do not include unnecessary line breaks or additional text beyond the specified format.

### Input Format:
You will receive the following data:

1. **Study examples:** A list of 12 study example pairs, formatted as:
   'example input 1: [[...]], example output 1: [[...]], ..., example input 12: [[...]], example output 12: [[...]]'
2. **Final input:** A single 10x10 list of lists on which you must apply the inferred transformation(s).
3. **Image input:** Additionally, you receive an image that visualizes the 12 study example pairs and the final input query.

Your goal is to determine the correct transformation and return the final output grid.

### Input:
Study examples:
example input 1: <2-dimensional array representing the input grid of example 1>
example output 1: <2-dimensional array representing the output grid of example 1>
...
example input 12: <2-dimensional array representing the input grid of example 12>
example output 12: <2-dimensional array representing the output grid of example 12>

Final input: <2-dimensional array representing the final query input grid>

Figure 15: The prompt used for the systematicity experiment when instructing LLMs in (text+image) mode. Text enclosed in sharp brackets $< \dots >$ is replaced by the actual examples. Additionally, the model is provided with the image in Figure 11.

