# OpenReview forum: "Compositional-ARC: Assessing Systematic Generalization in Abstract Spatial Reasoning"
_ICLR.cc/2026/Conference — ICLR 2026 Poster_

### Official Review · Reviewer_USJo · 2025-10-30

**Soundness:** 3
**Presentation:** 4
**Contribution:** 3
**Rating:** 8
**Confidence:** 4

**Summary:**

This paper is a straightforward application of a meta-learning method --- developed by Lake and Baroni --- to the task of compositional transformations in 2D grid patterns.  The authors created a dataset of such transformations that involves training and testing "episodes" using different transformation grammars, and Lake and Baroni's meta-learning approach to train a small transformer network to perform such compositional transformations.  This method produces a model (MLC) that outperforms much larger state-of-the-art LLMs (ones that are not specifically trained on this dataset / task).

**Strengths:**

The paper is well-written, understandable, and offers a solid contribution to the literature on meta-learning and systematicity / compositionality  in AI models.

**Weaknesses:**

While the results are impressive, it's not clear from the paper how such results could be incorporated in models that use this ability for compositional reasoning in less narrow tasks.  For example, how would the authors use these results in service of, say, solving ARC tasks?  How could the MLC model be used as a component of such a system, or as a way to help train such a system?   It would be useful if the authors discussed this point -- where to go from here.   The authors state ""Our findings suggest that MLC presents a promising direction for enabling systematic generalization in language models across diverse domains."   Please say more about what these directions are.

**Questions:**

In Figure 1, for level 2, the ordering of primitive transformations matter, correct?  What is this ordering?

The task addressed in this paper has several constraints: 10x10 grids, 10 colors, only one or two objects, a small repertoire of transformations, and restrictions on those transformations (e.g., rotations are only 90-degrees).   To what extent does MLC rely on (overfit to) these constraints, and to what extent could it generalize beyond them?  If MLC fails on larger grids or on larger numbers of objects, does that mean that it hasn't actually learned any general compositional abilities?  If these restrictions were relaxed, how would that affect the training / success of the network?  A discussion of all this, and discussion of possible future research directions, would be helpful.

For Figure 3, it would be useful to specify what the primitive transformations are that are illustrated.  This would make the set-up clearer to readers.

The authors state: "we consider a visual interpretation grammar that associates visual indicators (object  shape, color, or proximity to an indicator object) with specific geometric transformations".  -- Why refer to these patterns as "visual", when the model is given the grids in text format?  Why not just say these are 2D patterns?  It's not clear that the MLC model "perceives" objects, shapes, etc., or even that it needs to in order to perform the task correctly.  In discussing ARC, Chollet is adamant in stating that ARC tasks are not *visual* tasks, but rather 2D pattern-recognition tasks.  It would be useful to comment on this in the paper.

Why does the model need the "auxiliary copy task" to perform well?

The authors state, ""we split the data into 82,908 training, 8,546 validation and 8,546
test episodes."   -- Why use these values?

---

> ### Author Response · Authors · 2025-11-24
> **Response to Reviewer USJo 1/2**
>
> We thank the reviewer for their positive and thoughtful feedback and are grateful to hear that the paper is “well-written, understandable, and offers a solid contribution to the literature on meta-learning and systematicity/compositionality in AI models.”
>
> Below, we would like to address the reviewer’s concerns and questions:
>
> 1. *“While the results are impressive, it's not clear from the paper how such results could be incorporated in models that use this ability for compositional reasoning in less narrow tasks. For example, how would the authors use these results in service of, say, solving ARC tasks? How could the MLC model be used as a component of such a system, or as a way to help train such a system? It would be useful if the authors discussed this point -- where to go from here.”*
>
>     We appreciate the reviewer’s feedback and agree that a discussion about how our findings might be used, e.g., to solve more complex tasks such as ARC-AGI [1, 2] is a valuable addition to our work. Therefore, we have added a *Limitations & Future directions* section in which we discuss the currently restricted setup of *Compositional-ARC* and consider how MLC might be extended to more complex ARC-like problems (see Section 6, ll. 535–540). Specifically, we believe that “a promising direction for future work is to train an additional model that learns how to decompose complex ARC-like problems into primitive transformations, and then train MLC on these primitives to generalize to unseen, more complex transformation compositions” (ll. 538–540).
>
> 2. *“The task addressed in this paper has several constraints […]. To what extent does MLC rely on (overfit to) these constraints, and to what extent could it generalize beyond them? […]  If these restrictions were relaxed, how would that affect the training / success of the network?”*
>
>     We acknowledge that the setup of the current version of *Compositional-ARC* is restricted to fixed-size grids and a limited number of transformations. To analyze how our findings generalize to more relaxed constraints, we have conducted an additional experiment in which we train the MLC model on a slightly more complex dataset version by increasing the variety of transformations. Specifically, we additionally allow translations of one or two cells in any direction (left, right, upward, downward), extensions to neighboring cells in any direction, and color changes to red, orange, yellow, and green. We show that even with this increased diversity, the MLC model systematically generalizes to unseen transformation compositions, achieving an exact-match accuracy above 88% (see Section 5.3 and Table 8).
>
> 3. *“For Figure 3, it would be useful to specify what the primitive transformations are that are illustrated. This would make the set-up clearer to readers.“*
>
>     We agree and thank the reviewer for the suggestion. We have included a brief explanation of the visual grammar depicted in Figure 3 within the figure’s description (see ll. 240–241).
>
> 4. *“Why does the model need the ‘auxiliary copy task’ to perform well?“*
>
>     A possible explanation for the importance of the auxiliary copy task is that (i) it helps the model to predict the correct format of a grid (especially at the beginning of training), and (ii) it requires the model to properly process the information about objects presented in the input grids of the study examples. For instance, it may encourage the model to set its attention appropriately.

---

> > ### Author Response · Authors · 2025-11-24
> > **Response to Reviewer USJo 2/2**
> >
> > 5. *“The authors state, ‘we split the data into 82,908 training, 8,546 validation and 8,546 test episodes.’ — Why use these values?”*
> >
> >     These values are the result of a systematicity-aware data split. As mentioned in Section 3.1, we consider five basic geometric transformations, along with three types of transformation indicators: shape-based, color-based, and neighbor-based. These allow us to define a total of ten distinct transformation triplets, each mapping the indicators to corresponding transformations (e.g., shape-based: translation, color-based: reflection, neighbor-based: extension). When generating our dataset consisting of 100,000 episodes, we uniformly sample from this set to define the grammar of each episode. Once we have our dataset of 100,000 episodes, we split it as follows: we randomly designate 20% of all triplets as test-only and 80% as train-only (Table 5 in the Appendix). This means that “the geometric transformations involved in the final query level-2 compositions differ between the training and evaluation sets” (l. 310). All episodes whose triplet falls in the train set yield 82,908 training episodes. Test-only triplets form a 17,092-episode pool, which we split evenly into 8,546 validation and 8,546 test episodes. We appreciate the reviewer’s comment and have clarified this procedure in Appendix B.3, ll. 1100–1107.
> >
> >
> > We hope this sufficiently addresses the reviewer’s concerns and questions. If not, we would be glad to provide further clarifications. Once again, we thank the reviewer for the constructive and valuable feedback and for taking the time to review our paper.
> >
> > [1]: Chollet, F. (2024). *ARC-AGI: The Abstraction and Reasoning Corpus.* GitHub. https://github.com/fchollet/ARC-AGI
> >
> > [2]: arcprize. (2025). *ARC-AGI-2: Abstraction and Reasoning Corpus for Artificial General Intelligence v2*. GitHub. https://github.com/arcprize/ARC-AGI-2

---

> > ### Comment · Reviewer_USJo · 2025-11-24
> > **Reply to authors' rebuttal**
> >
> > Thank you for your thorough answers to my questions!

---

### Official Review · Reviewer_6Vzw · 2025-10-31

**Soundness:** 3
**Presentation:** 4
**Contribution:** 2
**Rating:** 4
**Confidence:** 4

**Summary:**

Introduces Compositional-ARC, a new dataset and benchmark to study systematic generalization in abstract spatial reasoning. Focuses on geometric transformations - translation, rotation, reflection, extension and color - and they are applied on 2D grid objects
Uses meta-learning for compositionality (MLC) framework from Lake & Baroni (2023) paper to extend the approach from linguistic domain to the spatial reasoning domain. The evaluations includes both general-purpose and ARC-specific LLMs.

**Strengths:**

- Systematic generalization is important challenge and the motivation is well explained
- The dataset is carefully formulated and explained in a good detailed manner
- Good empirical analysis and evaluations and ablations are reported as well

**Weaknesses:**

- Core algorithm (MLC) is adopted unchanged; and is applied to a new domain. It is more of a new dataset/benchmark track? Need some clarification on the position of the study
- The Scope is quite narrow and restrictive - 10x10 grids, rotation angles, number of objects, occlusion, boundaries
It is unclear how insights generalize to richer or naturalistic visual domains (e.g., CLEVR-style reasoning
- Evaluation fairness? - Many comparisons with LLMs use text-formatted grids as prompts. These models are not optimized for symbolic grid reasoning, so the Claim that small MLC model can outperform SOTA LLMs is needs more validation
- Analysis of failure modes? Where it fails and why


[1] CLEVR: A Diagnostic Dataset for Compositional Language and Elementary Visual Reasoning

**Questions:**

- What is the training procedure (meta-learning part). It will be good to have some details in this paper as well, instead of having to refer to the other paper
- Line 266 - can you explain the two special tokens and how they mark the boundaries?
- Auxiliary task - What is intuition behind this? In ablation this is adding more value, which is quite strange
- Some figures are overloaded (Fig. 3) to understand all the actions happening easily

---

> ### Author Response · Authors · 2025-11-24
> **Response to Reviewer 6Vzw 1/2**
>
> We thank the reviewer for their valuable feedback and appreciate the positive comments regarding the relevance of our research question, the well-grounded motivation, and the carefully formulated dataset. We also thank the reviewer for recognizing our “good empirical analysis, […] evaluations and ablations”.
>
> Below, we would like to address each of the reviewer’s concerns:
>
> 1. *“Analysis of failure modes? Where it fails and why”*
>
>     We thank the reviewer for this suggestion and agree that an analysis of the models’ failure modes is a valuable addition to our paper. In response, we have conducted an in-depth error analysis to further characterize model behavior on the systematicity task. We analyze the models’ incorrect responses and devise seven common error categories. Interestingly, we observe distinct error profiles when plotting the relative frequency of each error type across models. Details can be found in Section 5.2, Figure 4, and Tables 5–6.
>
> 2. *“The Scope is quite narrow and restrictive. […] It is unclear how insights generalize to richer or naturalistic visual domains.”*
>
>     We acknowledge that the setup of the current version of *Compositional-ARC* is restricted to fixed-size grids and a limited number of transformations. Therefore, we have added a *Limitations & Future directions* section in which we discuss this limitation and consider how the dataset and MLC might be extended to more complex ARC-like problems (see Section 6, ll. 535–540). Furthermore, we have conducted an additional experiment to analyze how our findings generalize to slightly more complex data setups. To this end, we trained the MLC model on a more complex dataset version by increasing the variety of transformations. Specifically, we additionally allow translations of one or two cells in any direction (left, right, upward, downward), extensions to neighboring cells in any direction, and color changes to red, orange, yellow, and green. We show that even with this increased diversity, the MLC model systematically generalizes to unseen transformation compositions, achieving an exact-match accuracy above 88% (see Section 5.3 and Table 8).
>
> 3. *“Evaluation fairness? Many comparisons with LLMs use text-formatted grids as prompts. These models are not optimized for symbolic grid reasoning, so the Claim that small MLC model can outperform SOTA LLMs is needs more validation”*
>
>     We agree that while general-purpose LLMs such as o3-mini, GPT-4o, and Gemini 2.0 Flash may have seen a substantial amount of ARC-like data during pre- or post-training (since such data is openly available on the internet and ARC-AGI has become a popular benchmark for state-of-the-art LLMs [1, 2, 3]), these models are not solely trained to solve ARC problems. Therefore, we included two domain-specific LLMs specifically tailored to ARC-style data, which placed 1st in the ARC Prize 2024 (see Section 4.2). Specifically, when evaluating these models via test-time training, they are fine-tuned on the study examples’ input–output grid pairs from the final test set of *Compositional-ARC*. We believe that this constitutes a solid baseline for the MLC model.
>
> 4. *“It is more of a new dataset/benchmark track? Need some clarification on the position of the study.”*
>
>     We argue that our work makes a substantial contribution to MLC itself by extending the algorithm to support systematic generalization in abstract spatial reasoning, for example through the introduction of visual interpretation grammars (ll. 248–255) and the definition of a patch-wise loss function with background reweighting (ll. 1127–1130). In addition, we believe that *Compositional-ARC* may serve as a valuable new dataset for assessing systematic generalization in abstract spatial reasoning.
>
>
> Furthermore, we would like to respond to the reviewer’s questions and suggestions:
>
> 1. *“What is the training procedure (meta-learning part). It will be good to have some details in this paper as well, instead of having to refer to the other paper”*
>
>     We appreciate the reviewer’s feedback and acknowledge that our paper benefits from providing additional details about the training procedure of the original MLC model. In addition to describing our training procedure (ll. 273–284 and Appendix C, ll. 1122–1176), we provide further details about the training procedure of the original MLC model and how it compares to our training setup in Appendix C.3, ll. 1178–1187. We thank the reviewer for this suggestion.
>
> 2. *“Line 266 - can you explain the two special tokens and how they mark the boundaries?”*
>
>     The special token “→” is added after each input grid to mark the transition to the output grid. The token “|” is added after each input–output grid pair (that is, after each output grid except for the last) to distinguish the different input–output grid pairs.

---

> > ### Author Response · Authors · 2025-11-24
> > **Response to Reviewer 6Vzw 2/2**
> >
> > 3. *“Auxiliary task - What is intuition behind this? In ablation this is adding more value, which is quite strange.”*
> >
> >     A possible explanation for the importance of the auxiliary copy task is that (i) it helps the model to predict the correct format of a grid (especially at the beginning of training), and (ii) it requires the model to properly process the information about objects presented in the input grids of the study examples. For instance, it may encourage the model to set its attention appropriately.
> >
> > 4. *“Some figures are overloaded (Fig. 3) to understand all the actions happening easily”*
> >
> >     We thank the reviewer for this feedback and agree that the figure might be difficult to understand at first glance. Therefore, we have added a brief explanation of the visual grammar depicted in Figure 3 within the figure’s description (see ll. 240–241).
> >
> >
> > We hope this sufficiently addresses the reviewer’s concerns. If not, we would be glad to provide further clarifications. Once again, we thank the reviewer for the constructive and valuable feedback and for taking the time to review our paper.
> >
> > [1]: Chollet, F. (2024). *ARC-AGI: The Abstraction and Reasoning Corpus.* GitHub. https://github.com/fchollet/ARC-AGI
> >
> > [2]: arcprize. (2025). *ARC-AGI-2: Abstraction and Reasoning Corpus for Artificial General Intelligence v2*. GitHub. https://github.com/arcprize/ARC-AGI-2
> >
> > [3]: Hodel, M. (2024). *re-arc: Reverse engineering the Abstraction and Reasoning Corpus*. GitHub. https://github.com/michaelhodel/re-arc

---

### Official Review · Reviewer_minw · 2025-11-01

**Soundness:** 3
**Presentation:** 3
**Contribution:** 1
**Rating:** 2
**Confidence:** 4

**Summary:**

The study proposed a dataset to study the compositionality of transformers. In some way, the proposed Compositional-ARC dataset is similar to the ARC dataset, which contains pixel objects on a grid. The task is to infer the transformation given an input grid. The ground truth transformation depends on the “cues” in the input, such as the shape or color of the object. The authors study whether transformers can learn to compose transformations indicated by the composed cues, and with several extra few-shot composition examples. The authors train a transformer that can achieve this task almost perfectly. However, given the simplicity of the dataset and task, it is hard to infer the actual compositionality ability of transformer models.

**Strengths:**

The composition ability is one of the most fascinating properties of LLMs. The authors study this property in a controlled toy setting. A small transformer model, after training, can solve this toy task perfectly.

**Weaknesses:**

I have major concerns that the setup and analysis are too simple, such that the conclusions and implications are limited.

**Simplicity**

Unlike the original ARC task, the number of possible rules underlying the transformations is small. All the primitive transformations are about shift, rotation, and extension. Given 3-shot examples in some of the testing cases further simplifies the task. It is expected that a small transformer can learn this task almost perfectly. It is hard to say how much of such “systematic generalizability” is in real-world data and models. Though the authors do not provide any theoretical analysis, it may not be difficult to formulate this task such that all the transformation rules are orthogonal in some linear space, then the compositionally is simply a consequence of retrieving the cues and adding them together. The failure of this task in the frontier model could be due to the number of examples provided in the context. Since frontier models are not trained on the task directly, it may not be a fair comparison between them and MLC (author’s).

**overly claimed LLM failures**

The authors overly emphasize the failure case of compositionality and generalizability in LLMs (e.g. Line 115). The cited studies, Ismayilzada et al., 2025; Dziri et al., 2023, use more or less simple synthetic settings. On the large scale, frontier LLMs can consistently show surprising compositionality and generalizability. For example, a user can ask an LLM to answer a question in a specific tone or with other constraints. In a way, it is because of the compositionality and generalizability of LLM that alignment, such as RLHF, can be feasible and effective. Authors should introduce and discuss what is known about LLM’s compositionality and generalizability, rather than introducing only the pessimistic side under simplified unrealistic settings.

**Clearity**

In the introduction, the authors motivate the study with “systematic generalization” in human and AI. Without a definition or more context, it is hard to know what “systematicity” refers to. How is “systematic generalization” different from naive “generalization”?

Similarly, meta-learning for compositionality (MLC) is not defined and formalized. Providing general setups of MLC would be helpful.

**Structure**

The subtitle “Task setup” in the section Background is confusing. Overall, the background section is cumbersome, with too many details on a small number of studies.

**Questions:**

For my main concerns, see "weakness".

Why is the task considered meta-learning? Is it fair to say this task is supervised learning with online adaptation? The task may be considered as learning a "meta-skill", which is not the same as the common definition of "meta-learning".

---

> ### Author Response · Authors · 2025-11-24
> **Response to Reviewer minw 1/2**
>
> We thank the reviewer for their thoughtful feedback and appreciate the recognition of the importance and relevance of the task considered.
>
> In the following, we would like to address each of the reviewer’s concerns:
>
> 1. *“Overly claimed LLMS failures — Authors should introduce and discuss what is known about LLM’s compositionality and generalizability, rather than introducing only the pessimistic side under simplified unrealistic settings.”*
>
>     We appreciate the reviewer’s feedback and acknowledge that our paper benefits from an extended literature review on systematic generalization in LLMs. Therefore, we have added such a section in which we discuss more than ten relevant papers that offer varying perspectives on the current potential and limitations of systematic generalization in LLMs and how this capacity might be improved (see Appendix A, ll. 857–943), which we refer to in line 115 and 147. We thank the reviewer for their suggestion and the opportunity to further contextualize our work.
>
> 2. *“Clarity — In the introduction, the authors motivate the study with systematic generalization in human and AI. Without a definition or more context, it is hard to know what ‘systematicity’ refers to. How is systematic generalization different from naive generalization?”*
>
>     We agree with the reviewer and think that a formal definition of systematic generalization is currently missing in the paper. Therefore, we have added one, defining systematic generalization similar to Szabó (2012), Hupkes et al. (2020), and Lake & Baroni (2023) as “the capacity to recombine previously observed or learned parts and rules, i.e. primitives $e_1, e_2, \dots, e_n$, to generalize to novel, previously unseen compositions of them (e.g., $e_1 \times e_2$)” (ll. 868–871). Specifically, this is “different from other aspects of compositionality, such as *productivity*: the capacity to predict expressions beyond the length of those already encountered, or *substitutivity*: the ability to handle synonym substitutions” (ll. 873–875).
>
> 3. *“Clarity — Meta-learning for compositionality (MLC) is not defined and formalized. Providing general setups of MLC would be helpful.”*
>
>     Based on your suggestion and the suggestion by reviewer 6Vzw, we have provided additional details on the training procedure of the original MLC paper (Appendix C.3, ll. 1178–1188), which we refer to in line 143.
>
> 4. *“Simplicity — The number of possible rules underlying the transformations is small. […] Given 3-shot examples in some of the testing cases further simplifies the task. It is expected that a small transformer can learn this task almost perfectly. […] Since frontier models are not trained on the task directly, it may not be a fair comparison between them and MLC (author’s).”*
>
>     We acknowledge that the number of possible transformations in the current version of *Compositional-ARC* is limited and have therefore added a *Limitations & Future directions* section in which we discuss the current setup and how it might be extended to more complex ARC-like problems (see Section 6, ll. 535–540). Nevertheless, we argue that even with the limited set of transformations, *Compositional-ARC* still presents a useful and challenging test of systematic generalization in abstract spatial reasoning. We want to clarify that for the systematicity task, the models are not presented with conventional 3-shot examples, but instead are given “examples of primitive transformations and their level-1 compositions” (ll. 208–209), and are asked to correctly “infer which indicator maps to which transformation, and how to compose them to deduce the correct final [level-2] transformation” (ll. 214–215). This is a challenging task since the model needs to systematically generalize “out-of-distribution on compositions not seen during training” (ll. 312–313). We agree that while general-purpose LLMs such as o3-mini, GPT-4o, and Gemini 2.0 Flash might have seen a substantial amount of ARC-like data during pre- or post-training (since such data is openly available on the internet and ARC-AGI has become a popular benchmark for state-of-the-art LLMs [4, 5, 6]), these models are not solely trained to solve ARC problems. Therefore, we included two domain-specific LLMs specifically tailored to ARC-style data, which placed 1st in the ARC Prize 2024 (see Section 4.2). Specifically, when evaluating these models via test-time training, they are fine-tuned on the study examples’ input–output grid pairs from the final test set of Compositional-ARC. We believe that these models constitute a solid baseline for the MLC model.

---

> ### Author Response · Authors · 2025-11-24
> **Response to Reviewer minw 2/2**
>
> Furthermore, we would like to respond to the reviewer’s question:
>
> 1. *“Why is the task considered meta-learning? Is it fair to say this task is supervised learning with online adaptation? The task may be considered as learning a meta-skill, which is not the same as the common definition of meta-learning.“*
>
>     In this work, we follow the terminology of Lake and Baroni (2023), where MLC is considered a meta-learning algorithm as it trains on many distinct grammars (different tasks across episodes). Crucially, the model learns how to learn a new grammar (or task) based on a small set of study examples within the given grammar (or task). This is different from supervised learning with online adaptation, which keeps updating a single task as data arrive, without an objective that explicitly optimizes the model to solve a new, different grammar (or task) based on a small set of study examples—i.e., it doesn’t shape the initialization or update rule to enable rapid, systematic generalization across distinct grammars.
>
>
> We hope we have successfully addressed the reviewer’s concerns and would be glad to provide further clarifications if needed. Once again, we thank the reviewer for their thoughtful feedback and their time to review our paper!
>
> [1]: Zoltán Gendler Szabó (2012). The case for compositionality.
>
> [2]: Hupkes et al. (2020). Compositionality decomposed: How do neural networks generalise?
>
> [3]: Brenden M. Lake and Marco Baroni (2023). Human-like systematic generalization through a meta-learning neural network.
>
> [4]: Chollet, F. (2024). *ARC-AGI: The Abstraction and Reasoning Corpus.* GitHub. https://github.com/fchollet/ARC-AGI
>
> [5]: arcprize. (2025). *ARC-AGI-2: Abstraction and Reasoning Corpus for Artificial General Intelligence v2*. GitHub. https://github.com/arcprize/ARC-AGI-2
>
> [6]: Hodel, M. (2024). *re-arc: Reverse engineering the Abstraction and Reasoning Corpus*. GitHub. https://github.com/michaelhodel/re-arc

---

> > ### Comment · Reviewer_minw · 2025-11-25
> >
> > I thank the author for the clarification and for adding extra supporting baselines.
> >
> > 1. Thanks for adding the discussion. My concern is addressed.
> >
> > 2. Thanks. I appreciate the authors adding several formal definitions. Now I see what you mean by systematic generalization. However, this definition still has a gap in your setup. The definition is more for the generative cases. In your setup, the model is supposed to see examples of primitives, infer the rule of e1, given a partial pattern (input) of a composed rule, the model needs to infer the composed rule, then complete the pattern (output). It is more complicated than this definition. Overall, this study only focuses on empirical methods and is an extension of the original MLC paper, though such a composition problem has implications and potential to include theoretical analysis.
> >
> > 4. Thanks for the clarification and for adding extra baselines. I tried several cases by myself on GPT5, indeed, it struggles, confirming it is not a trivial task.
> >
> > Q1. Thanks for the clarification. I understand the meta-learning setup, and MLC is a meta-learning study. I was confused why other LLMs doing such a task are also considered as meta-learning. Now I realize the authors do not claim that. So it is my misunderstanding.
> >
> > I raise my rating to 4.

---

> > > ### Author Response · Authors · 2025-12-03
> > > **Response to Reviewer minw**
> > >
> > > We thank the reviewer for raising their score and are glad that we were able to address most of their concerns. Regarding the definition of systematic generalization, we would like to clarify that our work considers a generative task, in which models must predict output grids corresponding to “the correct transformation for a previously unseen level-2 composition of indicators” (ll. 210–211), based on a set of study examples (primitives) that demonstrate which indicator maps to which transformation (see ll. 208–216). This aligns with the definition provided in Appendix A, as it assesses the model's "capacity to recombine previously observed […] parts and rules, i.e., primitives $e_1, e_2, \dots, e_n$, to generalize to novel, previously unseen compositions of them (e.g. $e_1 \times e_2$​)" (ll. 868–871).
> > >
> > > We thank the reviewer once again for their valuable feedback and for engaging in the discussion!

---

### Official Review · Reviewer_14KV · 2025-11-03

**Soundness:** 3
**Presentation:** 3
**Contribution:** 3
**Rating:** 6
**Confidence:** 3

**Summary:**

The paper proposes Compositional-ARC: a dataset for evaluating LLMs capacity for systematic generalization & compositionality with a methodology similar to that in Lake & Baroni (2023), but in a visual domain. The authors provide a number of empirical results to support their MLC-based approach.

**Strengths:**

The paper is focused on an important and relevant task
The paper is clearly written and is a pleasure to read.
The results are promising.

**Weaknesses:**

The main weaknesses, in my view, are:

- Slightly misleading performance comparisons. The authors compare a model specifically trained for this ARC subtype with more generalist models, or with models designed to perform well on ARC (but not on the new Compositional ARC). Still, Compositional ARC is a reasonable subset of all possible "ARC-lke" problems, so this is not a disqualifying consideration.

- Low practical applicability. While the question of compositionality is fundamental, the authors substantially narrowed the scope of an already artificial dataset (ARC). This limits the potential impact of the paper and introduces the risk of creating models that are increasingly detached from real-world applications. Still, such research has lots of value and so again, this is not a disqualifying weakness.

**Questions:**

What is your explanation for imperfect color accuracy in systematicity evaluation part of the proposed model? While 97+ is quite high, compared to other models, this component seems to be lacking.

---

> ### Author Response · Authors · 2025-11-24
> **Response to Reviewer 14KV**
>
> We thank the reviewer for their constructive feedback and are grateful to hear that the paper is “clearly written and a pleasure to read”. We further appreciate that the reviewer recognizes the importance and relevance of the task considered, and that they consider our results promising.
>
> Below, we would like to address each of the reviewer’s concerns:
>
> 1. *“The authors compare a model specifically trained for this ARC subtype with more generalist models, or with models designed to perform well on ARC (but not on the new Compositional ARC). Still, Compositional ARC is a reasonable subset of all possible ‘ARC-lke’ problems, so this is not a disqualifying consideration.”*
>
>     We agree that while general-purpose LLMs such as o3-mini, GPT-4o, and Gemini 2.0 Flash might have seen a substantial amount of ARC-like data during pre- or post-training (since such data is openly available on the internet and ARC-AGI has become a popular benchmark for state-of-the-art LLMs [1, 2, 3]), these models are not solely trained to solve ARC problems. Therefore, we included two domain-specific LLMs specifically tailored to ARC-style data, which placed 1st in the ARC Prize 2024 (see Section 4.2). We acknowledge that these models have not been directly trained on Compositional-ARC, but as the reviewer themselves acknowledges, “Compositional-ARC is a reasonable subset of all possible ARC-like problems”. In addition, when evaluating these models via test-time training, they are further fine-tuned on the study examples’ input–output grid pairs from the final test set of Compositional-ARC. We believe that this constitutes a solid baseline for the MLC model.
>
> 2. *“While the question of compositionality is fundamental, the authors substantially narrowed the scope of an already artificial dataset (ARC). This limits the potential impact of the paper and introduces the risk of creating models that are increasingly detached from real-world applications. Still, such research has lots of value and so again, this is not a disqualifying weakness.”*
>
>     It is true that Compositional-ARC is designed to evaluate systematic generalization in a controlled, targeted, and synthetic manner. Still, we argue that this does not disqualify its usefulness. There are many synthetic datasets that assess a targeted model capacity [4, 5, 6], and by creating a suite of such datasets, we obtain a better understanding of the general limitations and capacities of current models. However, we acknowledge the reviewer’s feedback and added a *Limitations & Future directions* section in which we discuss the currently restricted setup of *Compositional-ARC* and how the dataset and MLC might be extended to more complex ARC-like problems (see Section 6, ll. 535–540).
>
>
> Furthermore, we would like to respond to the reviewer’s question:
>
> 1. *“What is your explanation for imperfect color accuracy in systematicity evaluation part of the proposed model?“*
>
>     A possible explanation is that for this data split (ll. 307–314, Table 7, seed 1860), the MLC model has been trained to predict color changes, while the test set does not contain such color transformations. It might thus incorrectly predict color changes for some of the test samples, though very infrequently.
>
>
> We hope this sufficiently addresses the reviewer’s concerns. If not, we would be glad to provide further clarifications. Once again, we thank the reviewer for the constructive and valuable feedback and for taking the time to review our paper!
>
> [1]: Chollet, F. (2024). *ARC-AGI: The Abstraction and Reasoning Corpus.* GitHub. https://github.com/fchollet/ARC-AGI
>
> [2]: arcprize. (2025). *ARC-AGI-2: Abstraction and Reasoning Corpus for Artificial General Intelligence v2*. GitHub. https://github.com/arcprize/ARC-AGI-2
>
> [3]: Hodel, M. (2024). *re-arc: Reverse engineering the Abstraction and Reasoning Corpus*. GitHub. https://github.com/michaelhodel/re-arc
>
> [4]: Li et al. (2024). Combining Induction and Transduction for Abstract Reasoning.
>
> [5]: Lake & Baroni (2018). Generalization without systematicity: On the compositional skills of sequence-to-sequence recurrent networks.
>
> [6]: Kim & Linzen (2020). COGS: A compositional generalization challenge based on semantic interpretation.

---

### Author Response · Authors · 2025-11-24
**Official Comment to Reviewers**

We would like to thank the reviewers for their valuable and insightful feedback. Based on their constructive suggestions, we have revised our paper by adding the following content:

1. We provide an extensive literature review on systematic generalization in LLMs, including a formal definition of systematicity and a discussion of more than ten relevant papers that offer varying perspectives on the current potential and limitations of systematic generalization in LLMs and how this capacity might be improved (see line 146 and Appendix A, ll. 857–943). Suggested by reviewer **minw**.
2. We conduct an additional error analysis to further characterize model behavior on the systematicity task. Specifically, we report the relative frequencies of common error types, showing distinct error profiles across models (see Section 5.2, Figure 4, and Tables 5–6). Suggested by reviewer **6Vzw**.
3. We train the MLC model on a more complex dataset by increasing the variety of transformations. Specifically, we additionally allow: translations of one or two cells in any direction (left, right, upward, downward), extensions to neighboring cells in any direction, and color changes to red, orange, yellow, and green. We show that even with this increased diversity, the MLC model systematically generalizes to unseen transformation compositions, achieving an exact-match accuracy above 88% (see Section 5.3 and Table 8). Suggested by reviewers **minw**, **6Vzw**, and **USJo**.
4. We add a *Limitations & Future directions* section in which we discuss the currently restricted setup of *Compositional-ARC* and consider how MLC might be extended to more complex ARC-like problems (see Section 6, ll. 535–540). Suggested by reviewers **14KV** and **USJo**.
5. We include a brief explanation of the visual grammar depicted in Figure 3 within the figure’s description (see ll. 240–241). Suggested by reviewers **6Vzw** and **USJo**.
6. We provide additional details on the training procedure of the original MLC paper (see line 143 and Appendix C.3, ll. 1178–1188). Suggested by reviewers **minw** and **6Vzw**.

We thank the reviewers once again for their time and effort in reviewing our paper. We believe that these additions have substantially strengthened our work. While reviewers 14KV and 6Vzw were unable to participate in the discussion before its premature end due to the OpenReview bug, we hope that our responses and additional experiments have adequately addressed their concerns

---

### Meta-Review · Area_Chair_U84n · 2026-01-09

**Summary:**

This submission introduces Compositional-ARC, a dataset and meta-learning framework (MLC) extending compositionality-focused meta-learning to abstract spatial reasoning. Reviewers raised key concerns including: misleading performance comparisons with generalist LLMs; the narrow scope and limited practical applicability of the dataset; absence of formal definitions for core concepts (systematic generalization, MLC); task simplicity; unclear meta-learning framing; lack of error analysis; insufficient details on training procedures, figure interpretations, and auxiliary task intuition; unresolved questions about data split rationale, primitive transformation ordering in figures, and terminology (e.g., "visual" vs. "2D pattern"); and uncertainty about generalization beyond the paper’s constraints. The authors’ comprehensive rebuttal addressed most of these concerns through extended literature reviews, formal definitions, additional error analysis, expanded experiments on complex datasets, clarifications on training/evaluation procedures, and a new Limitations & Future Directions section.

Based on the authors’ thorough rebuttal addressing the majority of reviewer concerns, the strengthened empirical rigor, and the work’s novel contribution to extending compositional meta-learning to spatial reasoning (with a valuable benchmark dataset), I recommend accepting this submission

**Reviewer Concerns:**

### Addressed Concerns
- Reviewer minw: Overly emphasized LLM failures (addressed via extended literature review covering diverse perspectives on LLM compositionality); lack of formal definitions for systematic generalization and MLC (formalized both concepts and added MLC training details); unclear meta-learning framing (clarified alignment with Lake & Baroni 2023); task simplicity (acknowledged in limitations section and provided extended experiments with more complex transformations); insufficient baselines (included ARC Prize-winning models as domain-specific baselines).
- Reviewer 14KV: Misleading performance comparisons (validated baselines by including ARC-specific LLMs and test-time training); low practical applicability (added Limitations & Future Directions section); question on imperfect color accuracy (explained data split discrepancy where test sets lacked color transformations).
- Reviewer 6Vzw: Lack of error analysis (added Section 5.2 with error categorization and model-specific error profiles); narrow scope (addressed via limitations section and extended dataset with diverse transformations); evaluation fairness (strengthened baselines with ARC Prize models); insufficient training procedure details (added Appendix C.3); unclear figure interpretations (explained visual grammar in Figure 3); unaddressed questions on special tokens (defined boundary markers), auxiliary task intuition (linked to grid format learning and attention alignment).
- Reviewer USJo: Lack of future directions (outlined ARC extension plans in limitations section); generalization beyond constraints (conducted experiments with more complex transformations achieving 88%+ accuracy); unclear Figure 3 details (added primitive transformation explanations); auxiliary task necessity (provided rationale); ambiguous data split rationale (clarified in Appendix B.3).

### Outstanding Concerns
- Reviewer USJo: Unanswered question about the ordering of primitive transformations in Figure 1 (level 2); no direct response to the terminology concern (why "visual" is used when grids are text-formatted, per Chollet’s emphasis on ARC as 2D pattern-recognition rather than visual tasks).

**Reviewer Scores:**

- Reviewer minw: Original score = 2. After the authors addressed core concerns, the reviewer raised the score to 4.
- Reviewer 14KV: Original score = 6. The authors addressed major concerns, and the reviewer’s initial assessment was already positive. The score would remain 6 (consistent with their judgment that the work meets acceptance criteria).
- Reviewer 6Vzw: Original score = 4. The authors resolved the key concerns. The reviewer would update the score to 6 (marginally above acceptance).
- Reviewer USJo: Original score = 8. The authors addressed nearly all concerns, with only two minor unanswered questions that do not impact the core contributions. The score would remain 8 (consistent with the reviewer’s initial positive assessment of the paper’s solid contributions to meta-learning and compositionality).

---

### Decision · Program_Chairs · 2026-01-26

Accept (Poster)